# RvLLM: LLM Runtime Verification with Domain Knowledge

**Yedi Zhang**[*]
National University of Singapore
Singapore

**Sun Yi Emma**
National University of Singapore
Singapore

**Annabelle Lee Jia En**
National University of Singapore
Singapore

**Jin Song Dong**
National University of Singapore
Singapore

## Abstract

Large language models (LLMs) have emerged as a dominant AI paradigm due to their exceptional text understanding and generation capabilities. However, their tendency to generate inconsistent or erroneous outputs challenges their reliability, especially in high-stakes domains requiring accuracy and trustworthiness. Existing research primarily focuses on detecting and mitigating model misbehavior in general-purpose scenarios, often overlooking the potential of integrating domain-specific knowledge. In this work, we advance misbehavior detection by incorporating domain knowledge. The core idea is to design a general specification language that enables domain experts to customize domain-specific constraints in a lightweight and intuitive manner, supporting later runtime monitoring of LLM outputs. To achieve this, we design a novel specification language ESL and introduce a runtime verification framework RvLLM to validate LLM output against domain-specific constraints defined in ESL. RvLLM operates in two main stages: interpretation and reasoning. During interpretation, it derives interpretations of the specification based on the context, which then guide the reasoning process to identify inconsistencies. When new knowledge is derived, RvLLM issues a follow-up query to the LLM to further verify the consistency. We evaluate RvLLM on three representative tasks: violation detection against Singapore Rapid Transit Systems Act, numerical comparison, and inequality solving. Experimental results show that RvLLM effectively detects erroneous outputs across various LLMs in a lightweight and flexible manner. The results reveal that despite their impressive capabilities, LLMs remain prone to low-level errors due to a lack of formal guarantees during inference, and our framework offers a potential long-term solution by leveraging expert domain knowledge to rigorously and efficiently verify LLM outputs.

## 1 Introduction

Unlike rule-based systems [52] operating on predefined and deterministic rules, large language models (LLMs) [1, 23, 62, 43] learn data representation and processing automatically from the training datasets, achieving human-like or even superhuman performance, and have driven significant advancements in various practical applications [10, 33, 55, 44, 24]. However, their non-deterministic and unpredictable nature sometimes leads to inconsistent or erroneous outputs [3, 49], posing significant risks in safety-critical or knowledge-intensive domains. Recent studies [61, 5] have shown

---

[*]Corresponding author

39th Conference on Neural Information Processing Systems (NeurIPS 2025).

that such misbehavior in LLMs cannot be fully eliminated, underscoring the urgent need for dedicated verification and validation methodologies to enhance the reliability of LLM-generated outputs.

LLM testing [67, 27, 30, 68] primarily aims to establish comprehensive benchmarks to evaluate the overall model performance against domain-agnostic criteria–such as accuracy, coherence, and fairness–in alignment with the intended application. While these approaches effectively assess general behavior and reveal edge cases that may provoke unexpected responses, they are limited to predefined benchmarks and lack the specificity needed to address domain-specific assessment needs. LLM verification, instead, may serve as a complementary mechanism to LLM testing by providing formal guarantees on model behavior. Despite substantial progress over the past decade in neural network verification [8, 19, 29, 34, 59, 48, 65], existing methods exhibit notable limitations: they are primarily tailored to simpler model architectures–such as deep or convolutional networks–and classification tasks, struggling to scale to the complexity of modern AI models and their diverse functionalities. Furthermore, these approaches target general properties such as robustness and fairness, making them ill-suited for domain-specific properties. Therefore, developing a dedicated verification paradigm tailored to LLMs is crucial for ensuring certified and reliable outputs.

On the other hand, the decline of rule-based expert systems [25, 31, 22] in the late 20th century can be attributed to their inability to handle incomplete information and poor scalability in real-world complexity. Defining comprehensive rule sets for open domains has been proven infeasible [52, 28]. We argue that such rules can only be statistically approximated–a capability exemplified by LLMs. However, this approximation cannot substitute for explicit rule encoding, and learning-based models remain inherently unreliable, particularly in edge cases demanding strict adherence to domain-specific requirements. This highlights the need for verifying explicitly defined rules in a lightweight and incremental manner for practical deployment and maintainability.

**This paper.** Building on these foundations, we propose runtime verification with domain knowledge as a sustainable solution to ensuring reliable LLM behavior, motivated by two key insights: i) Runtime verification offers a lightweight yet rigorous means to bridge testing and formal verification by assessing system behavior against formal properties during execution; ii) Existing work primarily focuses on generic misbehavior detection while often overlooking the critical role of domain-specific expertise. Integrating such knowledge is crucial for handling edge cases and enhancing LLM reliability in specialized tasks, where domain knowledge can significantly improve performance.

To achieve this, we introduce RvLLM, an innovative runtime verification framework tailored to LLMs to ensure reliable outputs, particularly against axiomatic domain specifications. The idea is to automatically check whether the LLM-generated responses adhere to domain-specific constraints expressed in a simple, adaptable specification language. This simplicity and flexibility empower domain experts to encode their knowledge and formally define constraints that capture the expected behavior of LLMs in specialized applications. To support this, we introduce ESL, a general language tailored for encoding rule-based domain expertise. ESL integrates natural language with formal logic to ensure the adaptability to the natural language settings of LLMs while enabling the structured and formal representation of properties to verify. This design allows for the rigorous specification of domain-specific criteria across diverse LLM applications.

Figure 1 presents an overview of our proposed framework. Given an ESL specification provided by domain experts, RvLLM first extracts relevant information from the LLM's context and outputs to interpret the specification, generating a set of propositional formulae along with truth assignments for the corresponding propositions. Following a normalization step, these formulae are transformed into a standard form and subsequently validated using a forward chaining procedure. This process either detects inconsistencies due to logical contradictions or infers new knowledge. Once new knowledge is inferred, the query generation module issues targeted queries, evaluates the LLM responses, and detects inconsistencies when outputs contradict the previously inferred knowledge.

**Experimental results.** We evaluate the effectiveness of RvLLM through experiments on three representative tasks: violation detection against Singapore Rapid Transit Systems Act [51], numerical comparison, and inequality solving, using a diverse set of LLMs. The experimental results show that RvLLM significantly enhances the reliability of LLM output in these domain-specific tasks. In the violation detection task, employing RvLLM as a complementary mechanism increases the true positive rate (TPR) by 15.7% to 50.2% across various models. In the numerical comparison task, RvLLM detects nearly all errors in the LLM responses across 100 randomly generated questions. In the inequality solving task, RvLLM facilitates a systematic analysis of the step-by-step reasoning

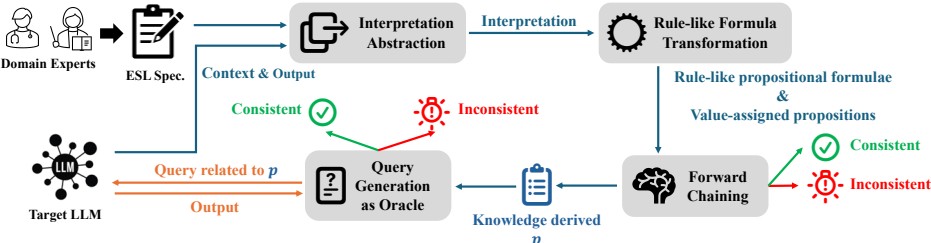

Figure 1: An overview of RvLLM. Given an LLM's context and outputs, RvLLM first extracts relevant propositions and translates the ESL specification into normalized propositional logic formulae. These are then processed through forward chaining to detect inconsistencies and infer new knowledge, which subsequently guides follow-up queries and consistency checks across successive LLM responses.

process from LLMs, effectively identifying its deviations from expert-defined specifications. These specifications are primarily grounded in fundamental algebraic inequality properties typically covered at the junior college level. Notably, despite the incomplete domain knowledge in the inequality solving task, RvLLM still achieves a TPR of 50% in detecting erroneous solutions. We argue that such a runtime verification approach can function as a critical safeguard for LLMs, particularly in rule-based or reasoning-intensive domains where adherence to specific constraints is essential. We further compare RvLLM with existing runtime hallucination-detection methods. Results show that RvLLM demonstrates superior performance by maintaining both high true positive and negative rates, whereas existing methods often exhibit a non-negligible trade-off between these metrics.

## 2 ESL: a simple way of specifying domain-specific properties

In this section, we introduce a general language, Expert Specification Language (ESL), which can be customized with domain-specified predicates by experts to impose behavioral constraints on LLMs.

### 2.1 Design guidance

In the following, we give our high-level requirements for designing a specification language that encapsulates LLM behavioral constraints from the perspective of domain experts. The core objective is to strike a balance between user-friendliness and expressiveness. We illustrate this with the following regulation from Singapore Rapid Transit System Act [51]:

> *"No person shall consume or attempt to consume any chewing gum or bubble gum while in or upon any part of the railway premises."*

**Accessibility for domain experts.** The primary goal of this specification language is to empower domain experts, even those with limited programming or formal methods experience, to effectively define and enforce constraints on the LLM's behavior. To balance expressiveness and user-friendliness, we base the specification rule on predicate logic [53, 20], restricted to prenex normal forms [53, 20] and consisting solely of universal quantifiers. We further constrain its quantifier-free part to a rule-like form (or implication), yielding a simplified variant of predicate logic expressible as $\forall \vec{x}.(f(\vec{x}) \Rightarrow g(\vec{x}))^2$. Specifically, both the left-hand side (LHS) and right-hand side (RHS) of the rule-like part are quantifier-free predicate formulae composed of predicates applied to terms that may include variables (e.g., $x$), constants (e.g., "bubble gum"), and functions of terms (e.g., $\sin(x)$). For example, $\mathsf{ChewGum}(x)$ is an atomic predicate with a single variable argument $x$, and $\mathsf{IsGreater}(\mathsf{Square}(x), 0)$ is also an atomic predicate with two arguments: the function term $\mathsf{Square}(x)$ and the constant term $0$.

**Ease of customization.** The specification language should be designed to support seamless customization, enabling adaptation to diverse domain-specific requirements. To facilitate this, we require each predicate in the specification to be associated with a natural language description that aligns with domain expertise. These descriptions allow customized predicates to remain clear, semantically grounded, and adaptable to specific application contexts. Additionally, built-in terms such as

---

[2]We use $\vec{x}$ instead of $x$ as there could be a sequence of variables.

arithmetic functions can be provided to streamline constraint formulation and reduce the need for extensive customization. For example, a domain expert can define their specialized atomic predicates $\mathsf{ChewGum}(x)$ and $\mathsf{InRailway}(x)$ associated with a natural language description as follows:

"$\mathsf{ChewGum}(x) :=$ a person $x$ consumes or attempts to consume chewing gum or bubble gum."
"$\mathsf{InRailway}(x) :=$ a person $x$ is in some part of railway premises."

**Efficient interpretation.** The language should also support an efficient interpretation procedure to obtain propositional formulae from the specification rule for subsequent logical verification. To this end, we remove the universal quantifier from the standard predicate formulas. For example, the regulation previously mentioned, typically expressed as $\forall x.(\mathsf{InRailway}(x) \Rightarrow \neg\mathsf{ChewGum}(x))$, will be simplified to $\mathsf{InRailway}(x) \Rightarrow \neg\mathsf{ChewGum}(x)$.

## 2.2 Formalization

In this section, we formalize our specification language, $\mathsf{ESL}$. We begin with the definition of Deductive Normal Form (DeNF), which serves as a foundation for the remainder of this paper.

**Definition 1.** *Given a propositional formula $\psi_1 \Rightarrow \psi_2$, if $\psi_1$ is a disjunctive norm form [53] and $\psi_2$ is a conjunctive norm form [53], then we call $\psi_1 \Rightarrow \psi_2$ is a deductive normal form (DeNF).*

**Definition 2.** *An $\mathsf{ESL}$ rule is a DeNF where each proposition is substituted by a predicate.*

An $\mathsf{ESL}$ specification $\mathcal{E}$ comprises three components: a variable set $\mathcal{V}$, a predicate set $\mathcal{P}$, and an $\mathsf{ESL}$ rule set $\mathcal{R}_\mathcal{E}$. For each predicate $f \in \mathcal{P}$, an associated natural language description is required. For instance, the following $\mathsf{ESL}$ specification encodes the regulation aforementioned in Section 2.1:

```
"Variables":    {"x"},
"Predicates":   {"ChewGum(x) := a person x consumes or attempts to consume chewing gum or bubble gum.",
                 "InRailway(x) := a person x is in some part of railway premises"},
"Rules":        {"InRailway(x) => not ChewGum(x)"}
```

**Interpretation of $\mathsf{ESL}$ rules**. An interpretation of an $\mathsf{ESL}$ rule $r$ assigns objects to variables (i.e., variable binding) in the predicate of $r$ and evaluates the truth value (True, False, or Unknown) of the resulting proposition after substitution. Note that we adopt the open-world assumption [17] to better accommodate the open-ended nature of LLMs. In this work, interpretations are derived from the context and LLM outputs, serving as the domain of discourse [26]. Now, consider the following contextual scenario:

*"In a crowded MRT train, Alex nervously chews gum to ease stress before an interview. Suddenly, the train jolts, and the gum flies out, landing on a stranger's shirt. Awkward glances turn into laughter as apologies spill out, diffusing tension in the confined space."*

The interpretation of the rule $\mathsf{InRailway}(x) \Rightarrow \neg\mathsf{ChewGum}(x)$ is $\{\mathsf{InRailway}(o_1) = \mathsf{True}, \mathsf{InRailway}(o_2) = \mathsf{True}, \mathsf{ChewGum}(o_1) = \mathsf{True}, \mathsf{ChewGum}(o_2) = \mathsf{Unknown}\}$, where $o_1 = $ 'Alex' and $o_2 = $ 'stranger'.

## 3 Methodologies of RvLLM

As illustrated in Figure 1, RvLLM conducts runtime verification through four stages: interpretation abstraction, rule normalization, forward chaining, and query generation. In this section, we detail the methodology of each stage and demonstrate its application using an example given in Figure 2.

### 3.1 Interpretation abstraction

Given a context and LLM outputs (the union denoted as $\sigma$) as the domain of discourse D and an $\mathsf{ESL}$ specification $\mathcal{E} = \langle \mathcal{V}, \mathcal{P}, \mathcal{R}_\mathcal{E} \rangle$, we first leverage a perception agent to obtain all possible objects from D and propositionalize as much as possible of the predicates defined in $\sigma$. We call this the *perception process*. Then, for each rule $r \in \mathcal{R}_\mathcal{E}$, we bind the variables in each predicate to all possible objects in a search-and-replace operation, to generate the propositional formulae set. Note that, given a

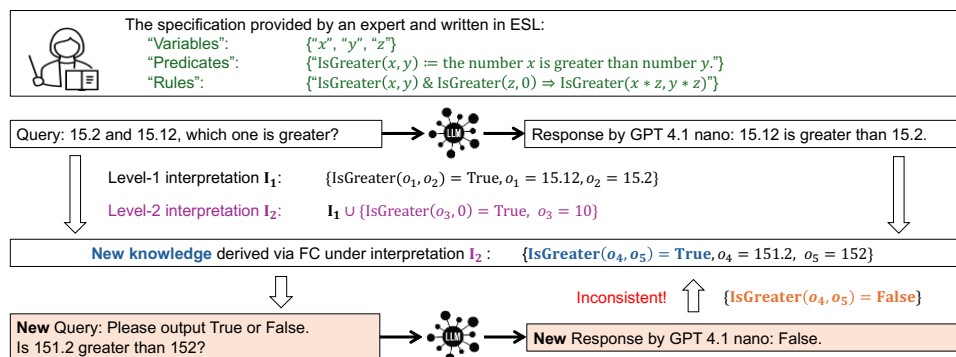

Figure 2: Runtime verification of GPT 4.1 nano by RvLLM for a number comparison task.

variable binding for a rule $r \in \mathcal{R}_\mathcal{E}$, the interpretation has two different results: i) all predicates in the LHS of $r$ are propositionalized successfully with assigned truth value under the binding, which we call a complete binding in this work, or ii) only part of predicates are propositionalized due to the insufficient information from the domain of discourse, which we call a partial binding.

Given this distinction, we introduce two levels of interpretation in this work: Level-1 and Level-2. Specifically, Level-1 interpretation disregards all the partial variable binding and operates only on the complete variable bindings. Given a partial binding, Level-2 interpretation, in contrast, employs a perception agent to instantiate all the variables that are not assigned with an object in the binding, in a way such that the resultant proposition returns True given the domain of discourse.

**Example 1.** Consider the example in Figure 2, which illustrates our approach using a simple numerical comparison question with a specification encoding a basic algebraic property of multiplication, the interpretation abstraction process first gets all possible objects and propositionalizes the predicate IsGreater as follows: $\{p_0 = \mathsf{IsGreater}(o_1, o_2) = \mathsf{True}, p_1 = \mathsf{IsGreater}(o_2, o_1) = \mathsf{False}\}$ with $o_1 = 15.2$ and $o_2 = 15.12$. Then, it can obtain all possible variable binding $\{\flat_1, \flat_2\}$ for the rule $\psi$ where $\flat_1 = \{x \mapsto o_1, y \mapsto o_2\}$ and $\flat_2\{x \mapsto o_2, y \mapsto o_1\}$. $\flat_1$ results in a formula $p_0 \wedge \mathsf{IsGreater}(z, 0) \Rightarrow \mathsf{IsGreater}(15.2 * z)$ for $\psi$, which is an partial binding. Similarly, $\flat_2$ is also partial. Consequently, for the Level-1 interpretation, no propositional formula is successfully generated, and the checking procedure exits with no inconsistency detected. While for the Level-2 interpretation, it successfully instantiates the variable $z$ for the partial bindings $\flat_1$ and $\flat_2$ with an object $o_3 = 10$, and obtain an additional proposition $p_2 = \mathsf{IsGreater}(o_3, 0) = \mathsf{True}$. Then, the updated bindings are: $\flat_1' = \{x \mapsto o_1, y \mapsto o_2, z \mapsto o_3\}, \flat_2' = \{x \mapsto o_2, y \mapsto o_1, z \mapsto o_3\}$. By applying $\flat_1'$, we can obtain a new proposition $p_3 = \mathsf{IsGreater}(o_1 * o_3, o_2 * o_3) = \mathsf{IsGreater}(15.2 * 10, 15.12 * 10)$ but with unknown truth value. Similarly, we obtain $p_4 = \mathsf{IsGreater}(o_2 * o_3, o_1 * o_3) = \mathsf{IsGreater}(15.12 * 10, 15.2 * 10) = \mathsf{Unknown}$ by $\flat_2'$. Finally, we generate two propositional formulae: $\psi_1 = p_0 \wedge p_2 \Rightarrow p_3$ via $\flat_1'$ and $\psi_2 = p_1 \wedge p_2 \Rightarrow p_4$ via $\flat_2'$ for the following component.

## 3.2 Rule-like propositional formula transformation

Given a DeNF formula $\psi = D_1 \vee D_2 \vee \cdots \vee D_m \Rightarrow C_1 \wedge C_2 \wedge \cdots \wedge C_n$, where each $D_i$ is a conjunction of literals $l_{i1}^D \wedge l_{i2}^D \wedge \cdots \wedge l_{im_i}^D$ and each $C_j$ is a disjunction of literals $l_{j1}^C \vee l_{j2}^C \vee \cdots \vee l_{jn_j}^C$, the formula transformation module is designed to reformulate $\psi$ into a set of propositional formulae $\Gamma_\psi$ in a rule-like form, i.e., an implication, such that the LHS of each formula in $\Gamma_\psi$ is a literal conjunction and the RHS is a single literal.

To achieve this, we first transform $\psi$ to a formula set $\Gamma_\psi' = \{D_i \Rightarrow C_j \mid i \in [m], j \in [n]\}$. Then, for each single formula $\psi_{i,j} = D_i \Rightarrow C_j = l_{i1}^D \wedge l_{i2}^D \wedge \cdots \wedge l_{im_i}^D \Rightarrow l_{j1}^C \vee l_{j2}^C \vee \cdots \vee l_{jn_j}^C$, we reformulate it as the implicant formula set $\Gamma_{\psi_{i,j}}'$, following the idea of unit propagation [63]:

$$\Gamma_{\psi_{i,j}}' = \{l_{i1}^D \wedge l_{i2}^D \wedge \cdots \wedge l_{im_i}^D \wedge (\bigwedge_{k \in [n_j] \wedge k \neq t} \neg l_{jk}^C) \Rightarrow l_{jt}^C \mid t \in [n_j]\}$$

Finally, given a DeNF $\psi$, we obtain the corresponding reformulated rule-like propositional formula set $\Gamma_\psi = \bigcup_{i \in [m], j \in [n]} \Gamma_{\psi_{i,j}}'$, and given a DeNF set $\mathcal{R}$, we use $\Gamma_\mathcal{R}$ to denote the union of all rule-like

form formulae transformed by each formula in $\mathcal{R}$, i.e., $\Gamma_{\mathcal{R}} = \bigcup_{\psi \in \mathcal{R}} \Gamma_\psi$. In the remainder of this paper, we use $\mathtt{Lit}_{\mathcal{R}}$ (resp. $\mathtt{Lit}_\psi$) to denote the set of all the literals that appeared in $\mathcal{R}$ (resp. $\psi$). The rules $\psi_1$ and $\psi_2$ obtained in Example 1 are already expressed in the rule-like form.

**Example 2.** Consider a DeNF $\psi = (a_1 \wedge b_1) \vee a_2 \Rightarrow (c_1 \vee d_1) \wedge c_2$. We first transform $\psi$ into an equivalent formula set $\Gamma'_\psi = \{\psi_{1,1}, \psi_{1,2}, \psi_{2,1}, \psi_{2,2}\}$, where $\psi_{1,1} = a_1 \wedge b_1 \Rightarrow c_1 \vee d_1$, $\psi_{1,2} = a_1 \wedge b_1 \Rightarrow c_2$, $\psi_{2,1} = a_2 \Rightarrow c_1 \vee d_1$, and $\psi_{2,2} = a_2 \Rightarrow c_2$. For each single formula in $\Gamma'_\psi$, we then reformulate it as follows: $\Gamma'_{\psi_{1,1}} = \{a_1 \wedge b_1 \wedge \neg c_1 \Rightarrow d_1, a_1 \wedge b_1 \wedge \neg d_1 \Rightarrow c_1\}$, $\Gamma'_{\psi_{1,2}} = \{a_1 \wedge b_1 \Rightarrow c_1\}$, $\Gamma'_{\psi_{2,1}} = \{a_1 \wedge \neg c_1 \Rightarrow d_1, a_1 \wedge \neg d_1 \Rightarrow c_1\}$, $\Gamma'_{\psi_{2,2}} = \{a_2 \wedge c_2\}$, and finally obtain the set of rule-like propositional formulae as $\Gamma_\psi = \Gamma'_{\psi_{1,1}} \cup \Gamma'_{\psi_{1,2}} \cup \Gamma'_{\psi_{2,1}} \cup \Gamma'_{\psi_{2,2}}$.

### 3.3 Forward chaining

Given a rule set $\Gamma_{\mathcal{R}}$ obtained as above with corresponding literal set as $\mathtt{Lit}_{\mathcal{R}}$ and a subset of literals $\mathtt{Lit} \subseteq \mathtt{Lit}_{\mathcal{R}}$ with predetermined truth values, the forward chaining (FC) procedure is designed to: i) verify consistency, and ii) infer truth values for the undefined literal $l \in \mathtt{Lit}_{\mathcal{R}} \backslash \mathtt{Lit}$. The truth value inference of undefined literals is operated based on the inference rule of modus ponens [14], which is closely aligned with the traditional forward chaining algorithm [52] utilized in rule-based systems. However, compared to the traditional FC algorithm, which requires each rule to be strictly a Horn Clause, a significant distinction between our algorithm and the traditional one is: we require that all negative literals appearing in $\mathtt{Lit}_{\mathcal{R}}$ are also included in the forward chaining graph. Such an extension enables us to detect the inconsistency by identifying truth values for undefined literals. For instance, consider a rule set $\{a \wedge b \Rightarrow c\}$ and literals defined as $a = b = \mathsf{True}$ and $c = \mathsf{False}$. In this case, the truth value of literal $c$ is *inconsistent* with the inference result derived from the rule set.

To achieve it, given a rule set $\Gamma_{\mathcal{R}}$ and an initially defined literals $\mathtt{Lit}$ with truth values, we first construct an initial graph $G$ through the following steps:

- Step 1 (Literal Nodes): Create a node for each literal that appears in the rule set $\Gamma_{\mathcal{R}}$.
- Step 2 (LHS Node): Create an LHS node for each rule-like propositional formula in $\Gamma_{\mathcal{R}}$.
- Step 3 (Edges): For each formula in $\Gamma_{\mathcal{R}}$, such as $l_1 \wedge \cdots \wedge l_n \Rightarrow l_{n+1} \in \Gamma_{\mathcal{R}}$, add a directed edge from each literal node $l_i$ ($i \in [n]$) to the corresponding LHS node, and add a directed edge from this LHS node to the literal node $l_{n+1}$.

Next, according to the initially defined literals, we mark all the literal nodes valued $\mathsf{True}$ as $\mathtt{Lit}^\uparrow$ and mark all the literal nodes valued $\mathsf{False}$ as $\mathtt{Lit}^\downarrow$, based on the truth value of literals from $\mathtt{Lit}$. Then, we perform forward chaining procedure on the graph $G$ and update $\mathtt{Lit}^\uparrow$ and $\mathtt{Lit}^\downarrow$ as follows until no more update can be done or an inconsistency is detected by $\mathtt{Lit}^\uparrow \cap \mathtt{Lit}^\downarrow \neq \emptyset$:

- Update of $\mathtt{Lit}^\uparrow$ and $\mathtt{Lit}^\downarrow$: For each LHS node encoding the rule-like formula (or implication) $l_1 \wedge \cdots \wedge l_n \Rightarrow l_{n+1}$, if $\{l_1, \ldots, l_n\} \subseteq \mathtt{Lit}^\uparrow$, then we have $\mathtt{Lit}^\uparrow = \mathtt{Lit}^\uparrow \cup l_{n+1}$ and $\mathtt{Lit}^\downarrow = \mathtt{Lit}^\downarrow \cup \neg l_{n+1}$.

**Example 3.** We now illustrate the graph construction and the corresponding FC procedure for the rule set $\Gamma_\psi = \{\psi_1, \psi_2\}$ under the Level-2 interpretation given in Example 1. We first obtain all the literals as $\mathtt{Lit}_{\mathcal{R}} = \{p_0, p_1, p_2, p_3, p_4\}$ with $\mathtt{Lit}^\uparrow = \{p_0, p_2\}$ and $\mathtt{Lit}^\downarrow = \{p_1\}$, construct the literal nodes correspondingly, and construct the LHS node set as $\{p_0 \& p_2, p_1 \& p_2\}$. Then, directed edges are added as follows: $p_0 \to p_0 \& p_2$, $p_2 \to p_0 \& p_2$, $p_0 \& p_2 \to p_3$, $p_1 \to p_1 \& p_2$, $p_2 \to p_1 \& p_2$, $p_1 \& p_2 \to p_4$, resulting an initial graph. Next, we check the logic consistency. For the LHS node $p_0 \& p_2$, since $\{p_0, p_2\} \subseteq \mathtt{Lit}^\uparrow$, we update $\mathtt{Lit}^\uparrow = \mathtt{Lit}^\uparrow \cup p_3 = \{p_0, p_2, p_3\}$ and $\mathtt{Lit}^\downarrow = \mathtt{Lit}^\downarrow \cup \neg p_3 = \{p_1, \neg p_3\}$, where "$p_3 = \mathsf{True}$" is the new knowledge derived. For the LHS node $p_1 \& p_2$, since $p_1 \notin \mathtt{Lit}^\uparrow$, there is no update on $\mathtt{Lit}^\uparrow$ and $\mathtt{Lit}^\uparrow$, hence no new knowledge inferred.

### 3.4 Query generation

Once $\mathsf{RvLLM}$ obtains the newly inferred knowledge, the query generation module generates a concrete query to the target LLM related to the knowledge, and requires the LLM to analyze its truth value. For the inferred knowledge $p_3 = \mathsf{IsGreater}(151.2, 152) = \mathsf{True}$ in Example 3, we generate the corresponding query as "*Is 151.2 greater than 152?*", as shown in Figure 2. Since the target LLM returns $\mathsf{False}$, an inconsistency is detected, indicating a diagnostic error in the target LLM.

**Clarification.** The soundness of this work—the ability to detect inconsistencies with respect to domain-specific constraints—depends on the accurate and faithful encoding of domain knowledge by the expert. Thus, the validity of RvLLM hinges on the alignment between the specification given in ESL and the underlying domain knowledge, and any misrepresentations in this encoding may compromise the soundness of the method.

## 4  Experiments

To evaluate the effectiveness of RvLLM, we apply it to the runtime verification of LLMs across three representative tasks that require domain-specific knowledge: violation detection against Singapore Rapid Transit Systems Act [51], numerical comparison, and inequality solving. For a comprehensive evaluation, we include both SOTA and non-SOTA LLMs as our benchmark, including Qwen 2.5 (max, plus, turbo, 7B, 14B, 32B, 72B), GPT (4.1, 4.1 mini, 4.1 nano), Gemini 2.0 (Flash, Flash Lite), and DeepSeek-V3. All experiments are conducted on a machine with an Intel (R) Xeon(R) w7-2475X processor. A dedicated guideline for designing interpretation abstraction prompts for the perception agent is provided in the appendices, along with additional experimental details and descriptions of the datasets and specifications used throughout the experiments.

To ensure a comprehensive evaluation, we also compare RvLLM with other runtime approaches, including SelfCheckGPT [47] and a prompt-based augmentation method that explicitly encodes domain-specific constraints within prompts. Due to the computational overhead and API costs associated with large-scale evaluations, we restrict our comparison to cost-efficient model variants. Additional experimental results and missing implementation details/justifications can be found in [64].

### 4.1  Case study 1: violation detection against Singapore Rapid Transit Systems Act

For this case study, we apply RvLLM under the Level-1 interpretation to evaluate the LLM's responses in detecting violations within contextual scenarios against the Singapore Rapid Transit Systems Act. In this capacity, RvLLM can be regarded as a complementary mechanism to enhance the LLM's capability in detecting the law violations in this study. We conduct experiments using contextual scenarios derived from [58], consisting of 304 cases in total–281 labeled as involving violations (unsafe) and 23 as not (safe). To incorporate relevant domain knowledge, we carefully encode 31 of the 53 regulations from Singapore Rapid Transit System Act into ESL specifications. It took a research scientist three days to interpret the law and develop corresponding specifications–a one-time effort applicable for verifying any LLM application against these regulations. In this study, the same model serves both as the target LLM and as the perception agent.

**Results.** Table 1 reports the violation detection results by various LLMs, with and without our framework as a complementary analysis method, where TPR and TNR denote the true positive rate and true negative rate for the violation detection, i.e., a violation is a positive example. The last column gives the average execution time for each scenario analysis. In the standalone LLM setting, a true positive is defined as a case where the model successfully identifies a violation in an unsafe scenario, and a true negative corresponds to a case where the model confirms the absence of violations in a safe scenario. In the combined LLM+RvLLM setting, a true positive is recorded if either the LLM or RvLLM detects a violation in an unsafe scenario, whereas a true negative is defined as a case that both the LLM and RvLLM determine the absence of any violation in a safe scenario. The results show that integrating RvLLM significantly enhances the TPR, improving the target LLM's capability to identify violations. However, we also observe that the integration of RvLLM can reduce the TNR in some models. This decline is primarily due to inaccuracies or incompleteness in the interpretation procedure by the perception agent. We also find that, among all evaluated models, DeepSeek-V3 achieves the highest TPR in both standalone and integrated settings, demonstrating superior detection capabilities in this domain-specific task and good performance on language understanding.

Figure 3 gives the comparison results of RvLLM against other baseline methods. We observe that SelfCheckGPT often yields the highest TPR gains, as it labels nearly all outputs as "hallucinations", resulting in near-zero TNR. This outcome is expected since SelfCheckGPT relies solely on output stability without leveraging domain knowledge, making it more suitable for open-domain generation. The prompt augmentation method (also known as LLM-as-a-judge) achieves TPR performance comparable to RvLLM but accompanied by a non-negligible TNR decrease. In contrast, RvLLM achieves a better balance between TPR and TNR, yielding more reliable and interpretable results.

Table 1: Violation detection results of LLMs with and without RvLLM against the Singapore Rapid Transit Systems Act [51]. All percentage values are reported rounded to one decimal place.

| Target LLMs | LLM | | LLM + RvLLM | | Performance of RvLLM | | Time (s) |
|---|---|---|---|---|---|---|---|
| | TPR | TNR | TPR | TNR | TPR | TNR | |
| Qwen max [62] | 56.2% | 91.3% | 86.1% | 91.3% | +29.9% | 0 | 3.00 |
| Qwen plus [62] | 58.4% | 1 | 81.9% | 1 | +23.5% | 0 | 5.32 |
| Qwen turbo [62] | 39.1% | 1 | 74.7% | 1 | +35.6% | 0 | 1.64 |
| Qwen 2.5 (7B) [62] | 16.0% | 1 | 61.2% | 87.0% | +45.2% | -13.0% | 2.28 |
| Qwen 2.5 (14B) [62] | 38.8% | 95.7% | 80.4% | 87.0% | +41.6% | -8.7% | 3.11 |
| Qwen 2.5 (32B) [62] | 56.6% | 95.7% | 87.5% | 95.7% | +31.0% | 0 | 2.55 |
| Qwen 2.5 (72B) [62] | 57.3% | 1 | 80.4% | 95.7% | +23.1% | -4.3% | 2.57 |
| GPT 4.1 [1] | 57.7% | 95.7% | 81.1% | 91.3% | +23.5% | -4.3% | 3.11 |
| GPT 4.1 mini [1] | 39.1% | 95.7% | 65.1% | 95.7% | +26.0% | 0 | 3.94 |
| GPT 4.1 nano [1] | 11.7% | 1 | 29.9% | 1 | +18.1% | 0 | 1.82 |
| Gemini 2.0 Flash [23] | 37.0% | 1 | 87.2% | 82.6% | +50.2% | -17.4% | 2.69 |
| Gemini 2.0 Flash Lite [23] | 54.8% | 95.7% | 79.0% | 95.7% | +24.2% | 0 | 1.72 |
| DeepSeek-V3 [43] | **78.3%** | 87.0% | **94.0%** | 82.6% | +15.7% | -4.3% | 15.84 |

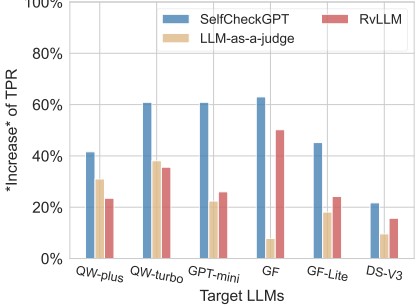

(a) The **increase** of TPR across models.

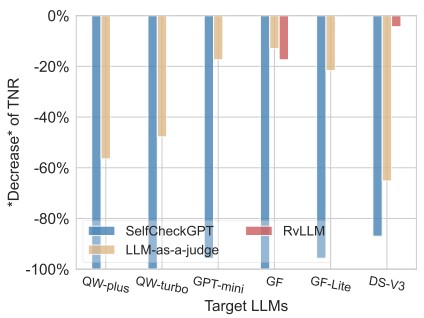

(b) The **decrease** of TNR across models.

Figure 3: Performance comparison of different methods in the violation detection task, where QW, GF/GF-Lite, and DS-V3 denote Qwen, Gemini 2.0 Flash/Flash Lite, and DeepSeek-V3, respectively.

## 4.2 Case study 2: numerical comparison problems

For this case study, we apply RvLLM under Level-2 interpretation to verify the numerical comparison results produced by various LLMs. It is important to note that RvLLM is inherently designed without mathematical solving capabilities and relies solely on logical inference. Without loss of generality, we randomly generate 100 questions following the guideline for increasing the likelihood of incorrect comparison results by LLMs. The specification file used here is the one shown in Figure 2. Again, we use the same model to serve both the target LLM and the perception agent.

**Results.** Table 2 gives the verification results by RvLLM on numerical comparison tasks across various LLMs. Columns Con. and Incon. report the number of cases where RvLLM detects no inconsistencies and where at least one inconsistency is detected, following the process in Figure 2. The values in parentheses in Column Con. represent cases where the target LLM produces an erroneous comparison result on the original comparison question, yet RvLLM fails to identify any inconsistencies–typically due to incomplete interpretation by the perception agents. In contrast, the values in parentheses in Column Incon. corresponds to cases where the target LLM returns a correct comparison result on the original question, but RvLLM indeed detects some inconsistencies. This usually arises from two main reasons: i) incomplete interpretation, and ii) the target LLM producing an incorrect diagnosis on newly-inferred knowledge. Overall, RvLLM effectively identifies erroneous diagnoses by the target LLM in most cases, with exceptions observed in only 3 cases for GPT 4.1 nano and mini. Among all tested models, Gemini 2.0 Flash Lite and DeepSeek-V3 show the weakest diagnostic performance, and Qwen 2.5 (7B) has the worst language understanding capabilities, failing all interpretation abstraction. Table 3 presents the performance comparison across methods. Consistent with the findings in case study 1, we find that RvLLM achieves the best balance, maintaining both high true positive and high true negative rates among all evaluated approaches.

Table 2: Verification results by RvLLM over various LLMs for numerical comparison tasks.

| Target LLMs | Con. | Incon. | Fail | Time (s) |
|---|---|---|---|---|
| Qwen max [62] | 93 | 7 | 0 | 5.03 |
| Qwen plus [62] | 99 | 1 | 0 | 7.01 |
| Qwen turbo [62] | 94 | 6(2) | 0 | 2.93 |
| Qwen 2.5 (7B) [62] | 0 | 0 | 100 | N/A |
| Qwen 2.5 (14B) [62] | 90 | 10(7) | 0 | 5.09 |
| Qwen 2.5 (32B) [62] | 98 | 2 | 0 | 4.98 |
| Qwen 2.5 (72B) [62] | 96 | 4 | 0 | 6.31 |
| GPT 4.1 [1] | 98 | 2 | 0 | 4.84 |
| GPT 4.1 mini [1] | 92(1) | 8 | 0 | 5.37 |
| GPT 4.1 nano [1] | 90(3) | 10 | 0 | 3.73 |
| Gemini 2.0 Flash [23] | 93 | 7 | 0 | 4.94 |
| Gemini 2.0 Flash Lite [23] | 52 | 44 | 4 | 4.35 |
| DeepSeek-V3 [43] | 69 | 31(5) | 0 | 30.87 |

Table 3: Performance comparison of different methods for numerical comparison tasks.

| Target LLMs | Methods | TPR | TNR |
|---|---|---|---|
| Qwen plus [62] | SelfCheckGPT | 0 | 96.7% |
| | LLM-as-a-judge | 100% | 0 |
| | RvLLM | 100% | 100% |
| GPT 4.1 mini [1] | SelfCheckGPT | 100% | 0 |
| | LLM-as-a-judge | 0 | 100% |
| | RvLLM | 88.9% | 100% |
| Gemini 2.0 Flash [23] | SelfCheckGPT | 87.5% | 0 |
| | LLM-as-a-judge | 0 | 100% |
| | RvLLM | 100% | 100% |

### 4.3 Case study 3: inequality solving problems

For this case study, we employ RvLLM under Level-1 interpretation to verify the step-by-step reasoning process of LLMs in inequality-solving tasks. We compile a dataset of 40 inequality questions sourced from A-Level H2 Mathematics examination papers [32] and carefully design three specifications to serve as domain knowledge that RvLLM uses for verification. These specifications consist of basic algebraic inequality properties taught at the junior college level, targeting three common LLM reasoning errors in inequality solving: incorrect factorization, flawed interval analysis, and omission of endpoint or critical point checks. Given the higher interpretation complexity in this case study and the strong language processing capabilities of DeepSeek-V3, we employ it as the perception agent for all LLMs evaluated and Qwen 2.5 (32B) as an alternative for comparison.

**Results.** Given the strong performance of RvLLM in detecting true positives, we focus the evaluation on the questions that the target LLMs answered incorrectly. The results are summarized in Table 4. Column 2 lists the total number of questions for which the target LLM produces incorrect solutions. Columns labeled DS-V3 and QW-32B (spanning Columns 3 to 11) show the number of cases where RvLLM successfully identifies inconsistencies based on the corresponding specifications using DeepSeek-V3 and Qwen 2.5 (32B) as the perception agent, respectively. Values in parentheses indicate false positives–cases where the detected inconsistency does not stem from an actual reasoning error but from inaccuracies of the perception agent during the interpretation abstraction process. Column GT gives the ground truth number of incorrect responses attributable to the rule specifications. It is important to note that we define only three specifications as domain knowledge for inequality-solving tasks, which is insufficient to cover all reasoning patterns required for such problems comprehensively. As a result, it is expected that RvLLM fails to detect a portion of the errors due to the lack of knowledge. The final two columns give the true positive rate of RvLLM with three specifications. We find that RvLLM achieves a TPR of up to 50% when using DeepSeek-V3 as the perception agent. This result underscores both RvLLM's potential to detect reasoning errors—even with limited domain knowledge—and the critical role of a robust perception agent in boosting overall performance. Although TPR remains below 50% for most models, they rise markedly when restricted to ground truth cases (Column GT), suggesting substantial gains are possible with richer domain knowledge in inequality solving.

## 5 Related work

**LLM testing.** Research in this area aims to develop comprehensive benchmarks and systematic evaluation frameworks to assess LLM across general attributes such as robustness, fairness, and factual consistency [45, 41, 57]. Robustness studies examine model resilience to adversarial input, paraphrasing, or noise [9, 56, 66], while fairness evaluations assess bias with respect to sensitive attributes like gender or race [39, 66]. Factual consistency detection, or hallucination detection, focuses on identifying instances in which LLMs produce plausible yet factually incorrect or unsupported claims [42, 38, 40]. Although large-scale benchmarks like HELM [41] and BIG-bench [54] offer task suites and metrics for performance evaluation, a critical challenge remains—the dynamic

Table 4: Verification results by RvLLM in inequality solving tasks with three specifications, defining rules related to factorization, interval analysis, and endpoints/critical points checking.

| Target LLMs | Incorrect | Factorization Error | | | Interval Error | | | Endpoints Error | | | TPR | |
|---|---|---|---|---|---|---|---|---|---|---|---|---|
| | | DS-V3 | QW-32B | GT | DS-V3 | QW-32B | GT | DS-V3 | QW-32B | GT | DS-V3 | QW-32B |
| Qwen max [62] | 12 | 1 | 1 | 1 | 1 | 0 | 1 | 2 | 2 | 3 | 33.3% | 25% |
| Qwen plus [62] | 10 | 1 | 1 | 2 | 0 | 0 | 0 | 0 | 1 | 2 | 10% | 20% |
| Qwen turbo [62] | 22 | 0 | 0 | 1 | 1 | 0 | 1 | 3 | 2 | 4 | 18.2% | 9.1% |
| Qwen 2.5 (7B) [62] | 25 | 2 | 1 | 3 | 7 | 2 | 7 | 3 | 2 | 5 | 48% | 20% |
| Qwen 2.5 (14B) [62] | 18 | 0 | 0 | 1 | 3 | 2(1) | 3 | 3(1) | 2 | 3 | 27.8% | 16.7% |
| Qwen 2.5 (32B) [62] | 16 | 1 | 0 | 1 | 3 | 2 | 4 | 4 | 2 | 5 | 50% | 25% |
| Qwen 2.5 (72B) [62] | 15 | 1 | 0 | 1 | 4 | 3 | 4 | 0 | 0 | 1 | 33.3% | 20% |
| GPT 4.1 [1] | 5 | 1 | 1(1) | 1 | 0 | 0 | 0 | 0 | 0 | 0 | 20% | 0 |
| GPT 4.1 mini [1] | 2 | 0 | 0 | 0 | 0 | 0 | 0 | 1 | 0 | 1 | 50% | 0 |
| GPT 4.1 nano [1] | 10 | 0 | 0 | 0 | 1 | 0 | 1 | 1 | 0 | 1 | 20% | 0 |
| Gemini 2.0 Flash [23] | 5 | 0 | 0 | 0 | 1 | 0 | 1 | 0 | 0 | 0 | 20% | 0 |
| Gemini 2.0 Flash Lite [23] | 6 | 1 | 1 | 2 | 0 | 0 | 0 | 0 | 0 | 1 | 16.7% | 16.7% |

and open-ended nature of real-world deployments complicates the construction of exhaustive and representative test suites.

**Runtime verification of LLMs.** Runtime verification [6, 21] is a dynamic analysis technique that ensures certified system behavior during execution by monitoring traces against formal specifications. Owing to its lightweight nature, runtime verification has inspired multiple efforts to verify LLM-generated outputs at runtime [13, 7, 47, 11, 37, 46, 36]. However, existing approaches center on open-domain settings, where specifications are often ambiguous and informal. For instance, SelfCheckGPT [47] and CHECKEMBED [7] operationalize specifications in terms of output stability. More recently, [12] proposes a fairness monitoring approach leveraging precisely defined properties in formal specifications. These specifications, informed by linear temporal logic [50] and its bounded metric variant [2], signal a shift towards formal methods for more accurate and dependable runtime verification of LLM behaviors. However, these works primarily focus on general properties and are not well-suited to domain-specific constraints for specialized tasks.

**Expert systems with NLP.** Rule-based expert systems [15, 16, 18] have effectively tackled domain-specific problems by coupling knowledge bases with inference engines to emulate expert reasoning for decades of years. To enhance usability, in the 70s, researchers integrated natural language processing (NLP), enabling interaction via simplified language, e.g., SHRDLU [60] in virtual environments and MedLEE [35] in clinical text analysis. These systems relied on symbolic parsing and controlled vocabularies but suffered from fragile parsing, limited lexical coverage, and ambiguity, often requiring constrained input or extensive user guidance. Nevertheless, integrating NLP with rule-based reasoning provides a key proof-of-concept for interactive, user-friendly AI, shaping later knowledge-based systems and natural language interfaces. Recently, the rise of LLMs has reinvigorated this paradigm. In this work, we employ a perception LLM to translate the target LLM's context and outputs into formal representations for backend reasoning. An improved version could explore other extensions of Horn clauses beyond those considered here to enhance expressiveness and completeness [4].

## 6 Conclusion

In this work, we present the first runtime verification framework RvLLM for LLMs, which allows the incorporation of domain knowledge. To support this, we design a general specification language, ESL, which enables domain experts to formally encode constraints in a lightweight yet expressive manner, supporting the rigorous verification of LLM behavior. We implement the proposed framework as a tool and evaluate its effectiveness on three representative tasks where domain knowledge plays a critical role. For each task, we develop tailored specifications grounded in relevant domain expertise. Experimental results indicate that RvLLM effectively leverages expert-defined domain knowledge to enable precise runtime verification of LLMs and strike a good balance between TPR and TNR, compared to existing runtime methods. However, despite its good performance, the current framework suffers from limited expressiveness by ESL, restricting its ability to encode more complex domain knowledge and behavioral constraints for LLMs. Future work will focus on enriching the specification language with additional operators to improve its expressiveness and generalization.

## 7 Acknowledgments

This work was partially funded by the Ministry of Education, Singapore, under its Academic Research Fund Tier 3 (MOET32020-0003). Any opinions, findings, conclusions, or recommendations expressed in this material are those of the author(s) and do not reflect the views of the Ministry of Education, Singapore.

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
