# OpenReview forum: "RvLLM: LLM Runtime Verification with Domain Knowledge"
_NeurIPS.cc/2025/Conference — NeurIPS 2025 poster_

### Official Review · Reviewer_6smZ · 2025-06-28

**Clarity:** 3
**Significance:** 2
**Originality:** 3
**Rating:** 5
**Confidence:** 4

**Summary:**

The paper proposes a method, RvLLM, for LLM output verification given a set of specifications. RvLLM uses a specification language (ESL) allowing domain experts to easily encode knowledge. At runtime, RvLLM interprets the specification given the LLM outputs to produce a set of propositions, then uses forward chaining to produce new knowledge and then query the LLM to detect inconsistencies. The experiments find that using RvLLM increases the TPR of detecting violations, while sometimes decreasing the FPR.

**Questions:**

- I'd suggest moving Related Work to the end before the conclusion for the flow of the paper
- If possible it would be nice to incorporate a lightweight example into Figure 1 to give a more more intuitive understanding at the start of the paper
- It would be interesting to see experiments where the LLM and RvLLM are different models. Particularly if a more lightweight model can effectively verify a larger model, it would decrease the overhead required to run this method.
- Another interesting experiment would be to examine the performance scaling as the number of specifications given increases -- e.g. at L362 with 3 specifications RvLLM achieves 52.2% TPR; at what point would we observe diminishing returns from adding more specifications? This sounds important for implementing this method in practice.

Minor nits:
- L203: Typo "bidding" should be "binding"
- Table 3: Columns are labeled DS-V1 instead of DS-V3
- Missing reference / baseline to Maieutic Prompting by Jung et al.

**Ethical Concerns:**

["NO or VERY MINOR ethics concerns only"]

**Final Justification:**

Thank you to the authors for the thorough response and the additional experiments! My questions and concerns have been addressed.

**Limitations:**

Yes

**Paper Formatting Concerns:**

No major formatting issues

**Quality:**

2

**Strengths And Weaknesses:**

Strengths:
- The method is well described and clearly formalized.
- ESL is a nice specification method for domain experts, usable for people who are not familiar with programming and formal logic.
- The application to transit rules violations is interesting and provides a good contrast between the mathematical domains and more natural rules.

Weaknesses:
- The paper is missing comparison to critical baselines that are mentioned in Related Work (SelfCheckGPT and CHECKEMBED). Also, simply using another LLM to verify the raw output, given just the input/output and the set of natural language rules (in the case of Singapore transit violations). Without these baseline to compare, it's difficult to judge how well the RvLLM method performs.
- The method requires a significant up-front time investment for each domain (3 days for the Singapore transit example) -- for more complex domains and extending to general behavior specification, this seems prohibitively expensive.
- L336-338: I think the conjecture about DeepSeek-V3 is a bit unfounded based on comparing just two domains.

---

> ### Author Rebuttal · Authors · 2025-07-31
>
> ## **Reply to Reviewer 6smZ**
>
> We thank the reviewer for the insightful comments and provide clarifications and additional experimental results below.
>
> ### **Weakness 1. Missing Baselines**
>
> Before presenting the comparison results with the baselines suggested by the reviewer, we would like to clarify the unique contribution of our work.
> This is the first work enabling runtime verification of LLMs with domain knowledge, where the knowledge can be incrementally encoded offline using a user-predefined specification language, ESL. **ESL combines both natural language and formal language, striking a balance between usability and precision.**
>
> Existing verification approaches, such as self-checking methods, rely mainly on the assumption that “stable outputs == non-hallucination”, which targets general-purpose settings. In contrast, **RvLLM verifies** LLM outputs against **explicit domain constraints** and can pinpoint **why** an output violates the specified knowledge.
>
> During the rebuttal period, we added comparison experiments with SelfCheckGPT and LLM-as-judge. For SelfCheckGPT, we evaluated all its embedding options. Due to the space, we currently only show the most suggested one (by SelfCheckGPT’s authors), i.e., SelfCheckGPT (NLI), with a sampling size of N=4 due to the token usages. For the LLM-as-judge method, we use deepseek-chat as the judge and encode natural language based domain constraints into its judging prompts. Note that we use deepseek-chat as the judge, as it has shown strong performance in our three case studies.
>
> Here we present our experiment results:
>
> #### **For Case Study 1: Violation detection against Singapore Rapid Transit Systems Act**
>
> | verified model  | Method | TPR Inc. vs Raw LLM outputs | TNR Dec. vs Raw LLM outputs|
> | - | - | - | - |
> | Qwen-plus  | SelfCheckGPT   | +41.6%  | -100%  |
> |   | LLM as judge | +30.9% | -56.5% |
> |   | **RvLLM**  | +23.5%  | -0%    |
> | Qwen-turbo  | SelfCheckGPT   | +60.9% | -100%  |
> |  | LLM as judge | +38.1% | -47.8% |
> |  | **RvLLM**  | +35.9% | -0%  |
> | gpt-4.1-mini  | SelfCheckGPT  | +60.9%  | -96%  |
> |   | LLM as judge    | +22.4% | -16%   |
> |   | **RvLLM**   | +26.0% | -3.4%  |
> | gemini-2.0-flash  | SelfCheckGPT  | +63.0%    | -100%  |
> |    | LLM as judge  | +7.8%     | -12%   |
> |   | **RvLLM**   | +50.81%   | -17%   |
> | gemini-2.0-flash-lite | SelfCheckGPT   | +45.2% | -96% |
> |    | LLM as judge    | +18.1%    | -20%   |
> |   | **RvLLM**   | +24.9%   | -0.3%   |
> | deepseek-chat     | SelfCheckGPT   | +21.7%   | -87.0%  |
> |   | LLM as judge  | +9.6% | -65.2%  |
> |    | **RvLLM**     | +15.7%       | - 4.3%   |
>
> We observe that  **RvLLM** achieves **more stable** performance across varied models, consistently maintaining a relatively low TNR drop while improving TPR high. In contrast, SelfCheckGPT often achieves the highest TPR increases simply because it tends to label nearly all outputs as “hallucinations” in our domain-specific settings, resulting in near-zero TNR. This behavior is expected, as SelfCheckGPT relies solely on output stability without incorporating domain knowledge. It does not encode or evaluate domain-specific rules, constraints, or context-sensitive reasoning, making it more suitable for open-domain generation tasks.
>
> As for LLM-as-judge method, we find it can achieve competitive TPR compared to RvLLM. However, its performance is also unreliable, with TPR gains typically accompanied by significant TNR drops. In contrast, RvLLM remains a better balance between TPR and TNR. This is because RvLLM leverages the LLM primarily for text understanding/reading, while delegating rule-based domain-related reasoning to an external reasoning engine, leading to more reliable and interpretable outcomes.
> We also include two additional comparison experiments corresponding to Case Studies 2 and 3.
>
> #### **For Case Study 2 and Case Study 3**:
>
> Due to the space limitation, please refer to the table results, which serve as a response to **Reviewer ywvD: Weakness 1**.
>
> Note that RvLLM’s relatively lower TPR in case study 3 stems from insufficient domain knowledge encoded. As more relevant knowledge is incorporated, TPR is expected to improve while still maintaining a high TNR.
>
> Regarding CheckEmbed. The reasons for not including it in the current comparison (during rebuttal) are outlined in our rebuttal W1 to Reviewer1: (1) high token cost, (2) limited time during the rebuttal period, (3) existing comparisons with SelfCheckGPT and llm-as-judge already reflect our effectiveness, and (4) CheckEmbed remains a preprint. That said, if the reviewer is still interested, we are also happy to extend our experiments with CheckEmbed in the coming weeks.
>
> ### **Weakness 2. Up-front time for each domain**
>
> We appreciate the reviewer’s concern regarding ESL authoring time. For the MRT domain, we used 31 official regulations, with the estimated 3 days reflecting total (non-continuous) working time, mainly spent interpreting legal texts rather than writing ESL rules. The annotator was a student familiar with ESL but not with MRT Law, representing a worst-case scenario. In familiar domains (e.g., math), writing rules (e.g., for inequalities) typically took under 1~10 minutes.
> Importantly, once written, ESL rules are reusable across models and tasks. Unlike red teaming or dataset-based validation, which require repeated efforts, **ESL offers a one-time setup that can be incrementally extended as needed.**
> Overall, we believe the upfront cost is reasonable and amortized, given ESL’s reusability, interpretability, and extensibility.
>
> ### **Weakness 3. Conjecture about deepseek-v3 in lines 336-338**
>
> We thank the reviewer for the feedback. We agree that the original conjecture may be too strong, given it is based on only two domains. In the revision, we will soften the presentation, present it as an empirical observation limited to the studied domains, and clarify that further investigation is needed to support any broader claims.
>
> ### **Question 1. Moving related work to the end**
>
> Thank you for the suggestion. We agree that moving the Related Work section before the conclusion improves the paper’s flow, and we will make this change in the revised version.
>
> ### **Question 2. A lightweight example**
>
> Thank you for the helpful suggestion. We agree that an early lightweight example would improve accessibility. In the revision, we will either integrate a simplified example into Figure 1 or move the existing running example (Figure 2) into the introduction to provide clearer intuition from the start.
>
> ### **Question 3. Exp. when LLM and RvLLM are different models.**
>
> In response to **Reviewer 9HFq**’s request (**Reviewer 9HFq: Weakness 3**), we conducted an additional experiment example to demonstrate RvLLM’s usability and breadth beyond the given three case studies. This new case involves alcohol consumption regulations in Singapore. We encoded 6 official rules and collected 200 real-world scenarios from the internet. In addition to standard evaluation (a similar setup to Case study 1), we also performed an ablation study treating the LLM and RvLLM as different models. The results are presented below.
> | verified LLM  | LLM as oracle used in RvLLM | LLM: TPR | LLM + RvLLM: TPR | TPR inc by RvLLM |
> | - | - | - | - | - |
> | qwen-max |  qwen-max | 67.5%    |   76.5%  |   +9% |
> |  |  qwen-turbo | 66.3%  |   71.4%  |  +5% |
> | gpt-4.1 | gpt-4.1 | 72%  |   82% |  +10% |
> | | gpt-4.1-nano  |   71.9% | 73.4% |  +1.5% |
> | gemini-2.0-flash | gemini-2.0-flash | 60.5% | 85%  |  +24.5% |
> | | gemini-2.0-flash-lite | 61.5%  | 71.5% |  +10% |
>
> The results show a noticeable drop in TPR when using a more lightweight model as the oracle LLM in RvLLM. This suggests that RvLLM’s effectiveness depends on the oracle model’s capability, as the accurate interpretation abstraction process requires a nuanced understanding of natural language. We will include these experimental results in the revised version.
>
> ### **Question 4. Long Tail Effect of Domain Knowledge in RvLLM**
>
> We thank the reviewer for raising this important, interesting, and practical question.
> While accurate abstraction combined with more domain knowledge generally improves TPR, we acknowledge the potential for diminishing returns as the number of specifications grows, especially when addressing long-tail or rare reasoning errors.
>
> To explore this, we conducted a detailed analysis on Case Study 3, examining performance as additional rules were added manually. Indeed, we identified three recurring error types that are challenging to capture under the current ESL design. Although the other typical error types were not fully experimentally analyzed, they also suggest diminishing returns from further rule additions. We will include this discussion and supporting analysis in the revised version.
>
> | verified LLM  | Incorrect Total Num | Factorization Error | Interval Error | Endpoints Error | Sign Handling Error | Simplification Error | Forgetting | Hallucination |
> | - | - | - | - | - | - | - | - | - |
> | qwen-max | 12 | 1 | 1 | 3 | 3 | 3 | 0 | 1 |
> | qwen-plus  | 10 | 2 | 0 | 2 | 1 | 1 | 3 | 1 |
> | qwen-turbo | 22 | 1 | 1 | 4 | 2 | 6 | 6 | 2 |
> | gpt-4.1 | 5  | 1 | 0 | 0 | 1 | 1 | 2 | 0 |
> | gpt-4.1-mini | 2  | 0 | 0 | 1 | 1 | 0 | 0 | 0 |
> | gpt-4.1-nano| 10 | 0 | 1 | 1 | 5 | 0 | 2 | 1 |
> | gemini-2.0-flash| 5  | 0 | 1 | 0 | 1 | 2 | 0 |1|
> | gemini-2.0-flash-lite|6|2|0|1|0|1|1|1|
>
> We find that "Endpoints Error," "Sign Handling Error," "Simplification Error," and "Forgetting" yield better returns under our benchmarks and verified LLM models, provided they can all be precisely encoded as domain constraints and embedded into RvLLM, even though the latter three remain difficult to represent accurately within the current ESL framework.
>
> Additional error types identified:
> - Sign handling error: Algebraic computation mistakes
> - Simplification error: Incorrect simplification of inequality
> - Forgetting: Incorrectly combining results across multiple cases

---

### Official Review · Reviewer_3yxj · 2025-07-02

**Clarity:** 3
**Significance:** 3
**Originality:** 3
**Rating:** 4
**Confidence:** 3

**Summary:**

This paper introduces RvLLM, a runtime verification framework for Large Language Models (LLMs), together with a lightweight Expert Specification Language (ESL). ESL enables domain experts to define task-specific rules in a hybrid syntax combining natural language and predicate logic. RvLLM operates in four stages: interpretation, rule normalization, forward chaining, and query generation. It dynamically detects inconsistencies between LLM outputs and expert-defined rules, optionally re-querying the model to verify critical facts. The framework is evaluated across three safety-critical scenarios—Singapore subway regulation checking, numerical comparison, and algebraic inequality solving—achieving substantial improvements in error detection (e.g., up to 50.2% increase in true positive rate for violation detection). Code and datasets are publicly released.

**Questions:**

Q1: Current examples mainly show single-premise, single-conclusion implication rules. Can ESL support more complex constructs, such as quantifiers, temporal logic, or coreference? Please discuss plans for syntax extension and usage in realistic regulation/finance corpora.

Q2: Please include misssing ablations.

**Ethical Concerns:**

["NO or VERY MINOR ethics concerns only"]

**Final Justification:**

Borderline Accept (4). The paper presents an innovative, well-articulated approach to LLM verification, with sound methodology and promising results. The weaknesses (efficiency, expressiveness) are not dealbreakers, but they do limit immediate applicability.

Authors are encouraged to:

- Extend ESL toward richer logic fragments.

- Improve runtime performance (e.g., caching, faster perception models).

- Apply RvLLM to broader, more realistic domains.

**Limitations:**

yes.

**Paper Formatting Concerns:**

No.

**Quality:**

2

**Strengths And Weaknesses:**

Strengths:

Clarity: Well-structured.

Significance: Runtime verification fills a critical gap for high-stakes applications like compliance and finance.

Originality: First to combine domain rule authoring and runtime validation in a unified LLM-centric framework; the query-to-refute strategy is particularly insightful.


Weaknesses:

Quality: Lacks quantitative comparison with recent verification or self-checking baselines such as SelfCheckGPT or CHECKEMBED.

Significance: Some cases take 15–30 seconds per output, limiting real-time use.

---

> ### Author Rebuttal · Authors · 2025-07-31
>
> ## **Reply to Reviewer 3yxj**
>
>
> We sincerely appreciate the reviewer's insightful feedback. We provide a detailed response along with more experimental results to address the raised concerns as follows.
>
>
> ### **Weakness 1. Quantitative comparison with recent baselines.**
>
> Before presenting the comparative results, we would like to clarify the novel contributions of our work. To our knowledge, this work represents the first framework enabling runtime verification of LLMs against domain-specific knowledge, where said knowledge can be: 1) **incrementally** encoded, 2) **in an offline manner** using a user-defined specification language ESL. ESL uniquely bridges natural language and formal specifications, achieving an optimal balance between usability and mathematical rigor.
>
>
> Existing verification approaches, such as self-checking-based methods, operate under the fundamental assumption that "output stability == non-hallucination". While these methods can detect the presence or likelihood of hallucinations in a more general setting, RvLLM offers distinct advantages:
>
> - Verification against domain-specific constraints
>
> - Precise identification of error sources
>
> - Explanation of why outputs violate encoded knowledge
>
>
> For a comprehensive comparison, we also conduct additional experiments with SelfCheckGPT and llm-as-judge:
>
> 1) SelfCheckGPT (all embedding options: BERTScore, N-gram, NLI, NQA, and Prompt-based). Sampling size fixed at N=4 due to computational constraints.
>
> 2) LLM-as-judge approach. We use DeepSeek-Chat as the judger, which demonstrated superior performance in our Case studies 1,2,3's results, and we also use it as a perception model in our Case study 3. When using deepseek-chat as the judge, we encode our natural language described domain constraints into the judge prompts.
>
>
> Here we present our experiment results:
>
> #### **For Case Study 1: Violation detection against Singapore Rapid Transit Systems Act**
>
> | verified model  | Method | TPR Inc. vs Raw LLM outputs | TNR Dec. vs Raw LLM outputs|
> | - | - | - | - |
> | Qwen-plus         | SelfCheckGPT   | +41.6%  | -100%  |
> |                   | LLM as judge    | +30.9%    | -56.5% |
> |                   | **RvLLM**              | +23.5%    | -0%    |
> | Qwen-turbo        | SelfCheckGPT   | +60.9%    | -100%  |
> |                   | LLM as judge     | +38.1%    | -47.8% |
> |                   | **RvLLM**              | +35.9%    | -0%    |
> | gpt-4.1-mini      | SelfCheckGPT   | +60.9%    | -96%  |
> |                   | LLM as judge    | +22.4%    | -16%   |
> |                   | **RvLLM**              | +26.0%    | -3.4%  |
> | gemini-2.0-flash  | SelfCheckGPT  | +63.0%    | -100%  |
> |                   | LLM as judge      | +7.8%     | -12%   |
> |                   | **RvLLM**              | +50.81%   | -17%   |
> | gemini-2.0-flash-lite    | SelfCheckGPT   | +45.2% | -96% |
> |                   | LLM as judge    | +18.1%    | -20%   |
> |                   | **RvLLM**                | +24.9%   | -0.3%   |
> | deepseek-chat     | SelfCheckGPT   | +21.7%   | -87.0%  |
> |                   | LLM as judge  | +9.6% | -65.2%  |
> |                   | **RvLLM**     | +15.7%       | - 4.3%   |
>
>
>
> We observe that RvLLM achieves more stable performance across varied models, consistently maintaining a relatively low TNR drop while improving TPR high. In contrast, SelfCheckGPT often achieves the highest TPR increases simply because it tends to label nearly all outputs as “hallucinations” in our domain-specific settings, resulting in near-zero TNR. This behavior is expected, as SelfCheckGPT relies solely on output stability without incorporating domain knowledge. It does not encode or evaluate domain-specific rules, constraints, or context-sensitive reasoning, making it more suitable for open-domain generation tasks.
>
>
> As for LLM-as-judge method, we find it can achieve competitive TPR compared to RvLLM. However, its performance is also unreliable, with TPR gains typically accompanied by significant TNR drops. In contrast, RvLLM remains a better balance between TPR and TNR. This is because RvLLM leverages the LLM primarily for text understanding/reading, while delegating rule-based domain-related reasoning to an external reasoning engine, leading to more reliable and interpretable outcomes.
>
>
>  We also include two additional comparison experiments corresponding to Case Studies 2 and 3.
>
>
> #### **For Case Study 2:**
>
> | verified model  | Method | TPR | TNR |
> | - | - | - | - |
> | Qwen-plus         | SelfCheckGPT    | 0  | 96.7%  |
> |                   | LLM as judge     | 0    | 0 |
> |                   | **RvLLM**                | 1    | 1  |
> | Qwen-turbo        | SelfCheckGPT   | 1    | 7.29%  |
> |                   | LLM as judge      | 0    | 0 |
> |                   | **RvLLM**                | 1    | 97.9%  |
> | gpt-4.1-mini      | SelfCheckGPT  | 1    |0  |
> |                   | LLM as judge      | 1    | 1  |
> |                   | **RvLLM**                | 88.9%    | 1 |
> | gemini-2.0-flash  | SelfCheckGPT   | 87.5%   | 0%  |
> |                   | LLM as judge  | 1     | 1   |
> |                   | **RvLLM**           | 1   | 1   |
> | gemini-2.0-flash-lite    | SelfCheckGPT  | 1 | 0 |
> |                   | LLM as judge    | 1    | 1  |
> |                   | **RvLLM**              | 1   | 1   |
> | deepseek-chat     | SelfCheckGPT   | 1   | 0  |
> |                   | LLM as judge   | 1 | 1  |
> |                   | **RvLLM**   | 1      | 93.2% |
>
>
> #### **For Case Study 3:**
>
>
> | verified model  | Method | TPR | TNR | Token Usage Per Task|
> | - | - | - | - | - |
> | Qwen-turbo        | SelfCheckGPT (NLI)   | 1    | 0  | 9850 |
> |                   | SelfCheckGPT (Prompt)   | 1    | 0  | 100527|
> |                   | LLM as judge      | 1    | 0 | 1466|
> |                   | **Ours**              | 18.2%    | 94.4%  | 4805 |
> | gpt-4.1-mini      | SelfCheckGPT (NLI)   | 1    | 2%  | 10373 |
> |                   | SelfCheckGPT (Prompt)   | 1    | 0  | 158504|
> |                   | LLM as judge      | 1    | 0 |  2479 |
> |                   | **Ours**              | 50%  | 84.2% | 5683 |
>
>
> Note that the reason why RvLLM achieves lower TPR is the insufficient domain knowledge embedded in the inequality-solving tasks. With more knowledge encoded, the TPR will increase while maintaining the TNR as quite high still.
>
>
>
> ### **Weakness 2.Efficiency Issue**
>
>
> Some cases take 15~30 seconds; the main reason behind this is the running time of the perception model, i.e., the oracle LLM used in RvLLM for interpretation abstraction. In the longest-running time cases, we use deepseek-chat as our interpretation abstraction perception model. By utilizing a more inference-efficient model, the running time can be largely reduced.
>
>
> ### **Question 1. Extension of ESL**
>
> We thank the reviewer for the thoughtful suggestion.
>
> Exactly, extension of ESL to more complex constructs is certainly possible and is also part of our ongoing exploration. Currently, our initial trials have shown that it is feasible to expand ESL expressiveness to support richer scenarios.
>
> However, we also note potential concerns: as the complexity of ESL specifications increases, the accuracy of interpretation and abstraction (particularly for more intricate scenarios) may degrade. Such a trade-off highlights the importance of carefully balancing expressiveness and interpretability-accuracy in future extensions. Maybe a runtime specification refinement process would be an alternative, which allows runtime refinement of the granularity of the specification.
>
> For future extensions to more enriched scenarios, a promising direction is to incorporate temporal operators into the specification language. Additionally, adapting the specification framework (e.g., TLA+) for smart contract verification offers a valuable and practical avenue. Ultimately, the trade-off between usability (i.e., user-friendliness and solving efficiency) and expressiveness remains a long-term research challenge.
>
>
> ### **Question 2. Missing Ablation.**
>
> Please see responses to **Weakness 1**.

---

> > ### Author Response · Authors · 2025-08-02
> >
> > Dear reviewer, we would like to kindly check whether we have addressed all your concerns. If you have any additional comments or suggestions, we would be happy to address/incorporate them during the remaining days of the rebuttal period.

---

> > ### Comment · Reviewer_3yxj · 2025-08-05
> >
> > Thanks for your replies. No issues remained.

---

> > > ### Comment · Area_Chair_nwKN · 2025-08-06
> > >
> > > Dear reviewer,
> > >
> > > I see that all the ratings remain the same as in your original review. Could you explain that given the author’s responses and that you said that no issues remain?

---

> > > > ### Comment · Reviewer_3yxj · 2025-08-06
> > > >
> > > > In their rebuttal, the authors provided additional experiments that effectively addressed my concerns. Nevertheless, had these results been presented in the original submission, I would have considered a score of 5. As such, I am keeping my current score of 4.

---

> > > > > ### Author Response · Authors · 2025-08-06
> > > > >
> > > > > Dear Reviewer, thank you for your follow-up. We sincerely assure you that the additional experimental results will be included in our revised version. We truly appreciate your thoughtful engagement and continued discussion.

---

### Official Review · Reviewer_9HFq · 2025-07-03

**Clarity:** 2
**Significance:** 2
**Originality:** 3
**Rating:** 3
**Confidence:** 2

**Summary:**

This paper introduces RvLLM, a runtime verification framework tailored for LLMs to ensure reliable outputs. Specifically, a general language called ESL is proposed for encoding rule-based domain expertise, and ESL integrates natural language and formal logic for answer verification. Experimental results conducted on three representative tasks demonstrate the effectiveness of the proposed method.

**Questions:**

- Are ESL specifications authored entirely by human experts, or is there an automated or semi-automated pipeline to assist their creation?
- Could you include additional examples or real-world scenarios to illustrate ESL’s expressiveness and breadth?

**Ethical Concerns:**

["NO or VERY MINOR ethics concerns only"]

**Final Justification:**

My concerns are largely resolved after discussing with authors. The only concern remains is the expert effort required to apply ESL across different domains, and I would like to see more detailed statistics of this. I will increase my score from 2->3.

**Limitations:**

Yes

**Quality:**

2

**Strengths And Weaknesses:**

Strengths:
- The introduction of ESL offers a lightweight, intuitive way for domain experts to encode rich, domain-specific constraints
- Experimental results on 3 representative tasks  demonstrate the effectiveness of the proposed method.

Weaknesses:
- The writing in this paper can be improved a lot. There are too many symbolic notations in the paper. In addition, introducing a concrete, running example in the Introduction would ground readers and improve accessibility.
- ESL’s coverage of domain knowledge is unproven. There is no evaluation demonstrating its reliability across varied or unforeseen cases. I am not sure if every domain constraint can represented using ESL.
- The evaluation uses only three carefully selected examples, raising concerns about the approach’s generalizability.

---

> ### Author Rebuttal · Authors · 2025-07-31
>
> ## **Reply to Reviewer 9HFq**
>
> We sincerely appreciate the reviewer's valuable feedback. Below, we provide detailed responses to each concern along with additional experimental evidence to substantiate our methodology.
>
> ### **Weakness 1. Writing issues**
>
> In the current version, we have intentionally limited symbolic notation to only those elements essential for validating the methodological rigor of our approach, which is sufficient for validating the solidity of the paper's methodology. We will incorporate additional explanatory text in the appendix.
>
> To enhance readability, we have included a concrete running example throughout the methodology section (Figure 2).
>
> In response to the reviewer's suggestion, we are happy to either: 1) Relocate this example to the introduction section for earlier understanding, or 2) Introduce an additional illustrative example to improve conceptual understanding.
>
>
> ### **Weakness 2.  Not sure if every domain constraint can be represented using ESL.**
>
> We understand the reviewer's concern that no evaluation demonstrating ESL's reliability across varied or unforeseen cases is provided in our paper. However, indeed, designing a specification logic or language to express every domain constraint is impractical. The logic (syntax and operators) should also be dedicated to the scenario complexity level and the questions of interest. Moreover, a higher complicated specification language will face : (1) hard in hand (user-friendliness consideration), (2) requiring a more computationally intensive solving algorithm, and limit the runtime efficiency.
>
> To illustrate the usefulness of ESL, we use daily scenarios in real life, including the Rapid Transit Systems Act, inequality solving problems, and number comparison tasks. To show the usability further, during the rebuttal period, we also include another real-world scenario about the Liquor consumption law in Singapore. By encoding related laws with ESL, RvLLM can successfully help LLMs do the violation detection tasks and monitor LLM's response related to these laws. We test on 200 liquor-consumption scenarios collected over the internet, and the TPRs are improved by up to 24.5% using RvLLM.
>
>
> Clarify: we enable the natural language description into our ESL syntax, hence giving a large extent of flexibility to the expert or users to define their domain constraints they would like to set limitations on the LLM behavior.
>
>
> ### **Weakness 3. Only three examples in our evaluations.**
>
> The reason why we use three examples is not because we can only fit these three, but because at the paper submission, we successfully collected expertise from three domains. Note that each additional benchmark/domain required dedicated domain expertise to encode the constraints, as well as the testing benchmark to evaluate.
> We would be happy to keep incorporating more domain-expertise incrementally (encoded by ESL),  collecting and enlarging our domain-expertise benchmark, i.e., exhaustive test cases given the domain-expertise settings.
>
> To eliminate the reviewer's concern, in this rebuttal period, we also collected Singapore law about "Liquor consumption" on the official site, and collected 200 alcohol consumption situations on the internet to act as our evaluation benchmark. We conduct similar experiments following the experiment setting as the Case Study 1. The experimental results are given in the following table.
>
> | verified model  | LLM: TPR | LLM + RvLLM: TPR |
> | - | - | - |
> | qwen-max              | 67.5%     |   76.5%   |
> | qwen-plus             | 66.3%     |   88.5%   |
> | qwen-turbo            | 92%       |   93.5%   |
> | gpt-4.1               | 72%       |   82% |
> | gpt-4.1-mini          | 66%   | 76.5% |
> | gpt-4.1-nano          | 20.1% | 37.7% |
> | gemini-2.0-flash      | 60.5% | 85%   |
> | gemini-2.0-flash-lite | 81%   | 81%   |
> | deepseek-chat         | 91.5% | 94.5% |
>
>
> ### **Question 1. Are ESL specification authored entirely by human experts?**
>
> ESL specifications are authored by human experts by default. However, to reduce their workload, experts may leverage modern AI tools, such as LLMs, to generate initial drafts of specifications, which are then required to be manually reviewed and refined. This would help avoid writing ESLs entirely from scratch, while ensuring correctness.
>
> It is important to emphasize that fully automated generation of formal specifications remains an open challenge: no existing method can guarantee both syntactic and semantic correctness of the fully-automated generated specifications. As such, human oversight remains essential in the specification creation process.
>
>
> Finally, we stress that the need for human-authored specifications is not a new challenge introduced in our work. Specification authoring has long been a fundamental requirement in the formal methods and model checking communities. Across all existing verification and model checking tools, specifications must be written by users to accurately encode the properties they wish to verify. Our work follows this established practice.
>
>
> ### **Question 2. Include additional examples or real-world scenarios to illustrate ESL's expressiveness and breadth.**
>
>
> Given the limited rebuttal time, we include an additional example, i.e., **Singapore alcohol drinking law** (used in real-world scenarios), to illustrate the real-world usage of RvLLM. Once again, we claim that we add this example is not because this is the only fitting scenario, but the one currently we can easily obtain the related domain expertise and collect benchmarks online in time. Also note that, our case study 1 and case study 3 are all based on domain expertise in real-world scenarios, and case study 2 comes from a notorious LLM hallucination on the number comparison task, which is also an interesting real-world problem when using LLM.
>
> Also, note that **we have provided detailed ESL encoding examples for all three real-world scenario case studies in the supplementary**.

---

> > ### Author Response · Authors · 2025-08-04
> >
> > Dear Reviewer, we truly appreciate your time and thoughtful consideration. We would like to kindly check whether we have addressed all your concerns. This paper is of great importance to us, and we have made a sincere effort to respond to each of your comments carefully. If possible, early feedback would be immensely helpful in allowing us to refine our responses and ensure that any remaining issues are fully resolved.

---

> > > ### Comment · Reviewer_9HFq · 2025-08-06
> > > **Response to Rebuttal**
> > >
> > > Thank you for your rebuttal. However, my concerns about the high costs and the expert effort required to apply ESL across different domains remain, so I will maintain my score.

---

> ### Author Response · Authors · 2025-08-06
>
> We sincerely thank the reviewer for their comments. However, we would appreciate clarification on the specific type of "cost" being referred to. Without this clarification, the conclusion that our method incurs "high cost" and requires "high expert effort" is difficult to evaluate and does not appear well-substantiated based on the evidence provided in our paper.
>
> ### **1. On the time cost**
>
> The execution time of our method can be as low as 2 seconds per verification task, with an average of less than 5 seconds across most settings. While the DeepSeek-v3 model takes longer, this overhead is due to the inference latency of the DeepSeek model itself, not our verification framework. We are confident that adopting a faster perception model would substantially reduce overall runtime. **RvLLM’s running time is comparable to that of LLM-as-judge methods and significantly more efficient than self-check-based approaches**, whereas RvLLM achieves substantially better performance in verifying outputs against domain-specific constraints.
>
> ### **2. On the ESL encoding effort**
>
> We acknowledge that ESL encoding requires a basic understanding of domain-specific constraints. However, this is not a limitation of our method, but of the verification task itself. **Any approach that aims to monitor/verify AI model outputs against specific domain requirements will inherently demand such properties encoding effort (as is evident in prior work on AI and CPS verification)**. That said, **our ESL framework has been specifically designed to minimize this encoding burden and make the encoding process a one-time effort**. For example, encoding constraints for a math problem typically takes only 1–10 minutes, and in our new case study on alcohol consumption regulations, all rules were encoded in under 5 minutes (This efficiency comes from our earlier experience encoding Rapid Transit Systems Act regulations using ESL, particularly in legal domains).
>
> **We also encourage the reviewer to refer to the example provided in the paper (e.g., Figure 2)**. For instance, we define the predicate as "IsGreater(x, y) := x is greater than y", and the rule  "IsGreater(x, y) & IsGreater(z, 0) -> IsGreater(x * z, y * z)" for the numeric comparison task. We believe these constructs are intuitive and accessible, even for users without a formal methods background.
>
> Furthermore, manual constraint specification is a longstanding and standard practice in the formal methods community. ESL improves upon this by offering a more user-friendly syntax, logical operators, and natural language integration, striking a practical balance between expressiveness and usability.
>
> ### **3. On token cost**
>
> Our method indeed uses a higher, yet comparable, number of tokens compared to LLM-as-judge approaches (when accessed via APIs), but achieves much better performance in domain-specific verification tasks. Moreover, compared to self-check-based methods, our approach both requires fewer tokens and delivers superior verification performance.
>
>
> We hope this response helps clarify our position and **resolve any potential misunderstandings regarding the (runtime) verification background**. In light of this, we would greatly appreciate it if the reviewer could kindly provide a well-justified and concrete evaluation based on the updated clarifications.

---

> > ### Comment · Reviewer_9HFq · 2025-08-08
> > **Response to Authors**
> >
> > Dear Authors,
> >
> > Thank you for your detailed justification. My concern is mainly on (2) the ESL encoding effort.
> >
> > You mentioned in the resonse
> > > For example, encoding constraints for a math problem typically takes only 1–10 minutes, and in our new case study on alcohol consumption regulations, all rules were encoded in under 5 minutes (This efficiency comes from our earlier experience encoding Rapid Transit Systems Act regulations using ESL, particularly in legal domains).
> >
> > However, these timings may reflect the **authors’ familiarity** with the language. Please consider the learning curve for domain experts who are not already proficient in ESL. I suggest adding evidences, e.g., a user study with non-author annotators in the revised version.
> >
> > I will lower my confidence and would like to hear authors' further justification.

---

> ### Author Response · Authors · 2025-08-08
>
> Dear Reviewer, we sincerely thank you for your comments and continued engagement.
>
> We fully understand your concerns regarding the potential learning curve for domain experts unfamiliar with ESL. However, we would like to clarify that conducting a formal user study, especially involving legal professionals, is non-trivial, given the difficulty of recruiting participants with both domain expertise and willingness to engage in such studies. That said, we agree this is a valuable direction and would be happy to pursue such a study into the revision if it is deemed essential for broader adoption. We genuinely appreciate your suggestion.
>
>
> In the revised version, we will include a more detailed explanation of ESL’s syntax along with additional illustrative examples that highlight its lightweight and intuitive design. **As you kindly noted in your review’s Strength section**—“ESL offers a lightweight, intuitive way for domain experts to encode rich, domain-specific constraints”—we believe that domain experts with basic familiarity with discrete mathematics (e.g., Boolean operators such as And, Or, Implies, Disjunctive Normal Form, and Conjunctive Normal Form) can readily grasp ESL, especially with the support of examples and documentation that we will release publicly in future.
>
> To further clarify its lightweightness and usage, ESL can be seen as a fragment of existing widely-used specification languages, like Signal Temporal Logic (STL), (if the predicate element in ESL is replaced by an atomic proposition). Here we give more syntax comparisons:
>
> - STL syntax: $ \varphi $ :=  $ \top $ | $ \neg \varphi $  | $ \varphi \wedge \varphi $ | $ \varphi \vee \varphi $ | $ F_{[a, b]} \varphi $ | $ G_{[a, b]} \varphi $ | $ U_{[a,b]} \varphi $
>
> - ESL syntax: $ \varphi := \varphi_1 ⇒ \varphi_2 $, where
>
>    - $ \varphi_1 $ is a Disjunctive normal form (DNF)
>    - $ \varphi_2 $ is a Conjunctive normal form (CNF)
>    - Each atomic proposition in DNF and CNF is a user-defined predicate.
>
> This ESL's syntax, embedded with predicates defined in natural language, strikes a balance between expressiveness and ease of use. For example, in the alcohol regulation case study, the following rule from Singapore's Liquor Control Act:
>
> ``If requested by the Licensing Officer or a police officer (whether or not an authorised officer), produce the licensee’s liquor licence for inspection.``
>
> can be expressed in ESL as:
>
> **Predicate**:
> - AskLicense(x) := person x is asked to produce or show a liquor license by a licensing officer or police officer (whether or not authorized)
> - ProduceOrShow(x) := x successfully produces or shows a valid liquor license upon request
>
> **Rule**:
> - AskLicense(x) $ ⇒ $ ProduceOrShow(x)
>
> We will include detailed usage guidelines in our upcoming open-source release to support users further.
>
> In summary, we believe that the ESL encoding effort remains relatively low for its intended flexible use cases—particularly within the verification community—and should not overshadow the broader contributions of our work. We sincerely hope this clarification helps address the reviewer's concerns.

---

> > ### Comment · Reviewer_9HFq · 2025-08-09
> > **Thank You for Your Further Justification**
> >
> > Thank you for your further justification. I will increase my final score.

---

> > > ### Author Response · Authors · 2025-08-09
> > >
> > > Dear reviewer, we sincerely thank you for your follow-up and hope our response fully addresses your concerns.

---

### Official Review · Reviewer_ywvD · 2025-07-03

**Clarity:** 3
**Significance:** 3
**Originality:** 3
**Rating:** 5
**Confidence:** 3

**Summary:**

The paper introduces RvLLM, a runtime verification framework that leverages domain-specific knowledge to validate large language model (LLM) outputs. The authors design a lightweight Expert Specification Language (ESL), enabling experts to express constraints as simple pairs (rule to constraint). RvLLM’s pipeline comprises four stages: interpretation abstraction, rule normalisation, forward chaining, and query generation, to detect logical inconsistencies and then issue follow-up queries when contradictions arise. The framework is evaluated on three representative tasks, including violation detection against the Singapore Rapid Transit Systems Act, numerical comparison, and inequality solving, demonstrating substantial improvements in error detection across a variety of LLMs.

**Questions:**

1. One thing unclear to me is: Regarding the term verification, does RvLLM have the mathematical guarantee like formal verification, i.e., regarding soundness and completeness, or is it more like validation/detection but combining with domain knowledge?
2. What if the domain knowledge is insufficient?

**Ethical Concerns:**

["NO or VERY MINOR ethics concerns only"]

**Final Justification:**

I maintain my original score as discussed with the authors/AC. The authors are encouraged to use a more suitable term, e.g., “runtime validation”, or state very clearly in the revised version that the results are subject to the assumption of certain domain knowledge, and highlight the potential risk of unsoundness; also, the limitations of the generalisation of ESL should be further explained in the paper.

**Limitations:**

Yes

**Quality:**

3

**Strengths And Weaknesses:**

Strengths:

1. By combining formally defined ESL rules with lightweight runtime checks, RvLLM bridges the gap between general-purpose LLM error detection and domain-specific validation, like the combination of neural-symbolic with LLM, which is interesting.
2. RvLLM was tested on legal compliance, numerical comparison, and inequality solving with a diverse set of state-of-the-art and smaller LLMs.
3. The final stage, 'Query Generation', would be more valuable  if it could be used to analyse the inconsistency that comes with some explanation

Weaknesses:
1. Evaluation compares only to raw LLM outputs, omitting other runtime or hallucination-detection methods that the authors mentioned in the paper, such as SelfCheckGPT or CHECKEMBED,
2. Although per-scenario times are reported (Table 1), the paper does not analyse throughput or resource usage for large-scale deployments
3.  ESL currently handles simple logical and arithmetic predicates; it is a bit unclear to me of its ability to encode temporal, probabilistic, or fuzzy constraints.

---

> ### Author Rebuttal · Authors · 2025-07-31
>
> ## **Reply to Reviewer ywvD**
>
> We sincerely appreciate the reviewer’s thoughtful feedback and constructive engagement with our work. In the following sections, we address each of the raised concerns and questions in detail.
>
> ### **Weakness W.1: Comparison to other baseline methods**
>
> To ensure a comprehensive evaluation, we conduct additional experiments covering all three case studies to compare our tools with 1) SelfCheckGPT (with NLI option and sampling time N = 4) and 2) an llm-as-judge (deepseek-chat) baseline method, where we explicitly encode domain-specific natural language constraints into the prompt. Due to the substantial computational overhead and API costs associated with large-scale evaluation, we opt for a cost-efficient model variant (cheaper api price). The comparative results are presented below. Note that we also report the token usage for case study 3, given the complexity and scalability of the task.
>
> #### **For Case Study 1: Violation detection against Singapore Rapid Transit Systems Act**
>
> | verified model  | Method | TPR Inc. vs Raw LLM outputs | TNR Dec. vs Raw LLM outputs|
> | - | - | - | - |
> | Qwen-plus  | SelfCheckGPT   | +41.6%  | -100%  |
> |  | LLM as judge  | +30.9%    | -56.5% |
> |  | **RvLLM**    | +23.5%    | -0%    |
> | Qwen-turbo   | SelfCheckGPT   | +60.9%    | -100%  |
> |   | LLM as judge     | +38.1%    | -47.8% |
> |     | **RvLLM**    | +35.9%    | -0%    |
> | gpt-4.1-mini      | SelfCheckGPT   | +60.9%    | -96%  |
> |    | LLM as judge    | +22.4%    | -16%   |
> |   | **RvLLM**  | +26.0%    | -3.4%  |
> | gemini-2.0-flash  | SelfCheckGPT  | +63.0%    | -100%  |
> |   | LLM as judge      | +7.8%     | -12%   |
> |    | **RvLLM**   | +50.81%   | -17%   |
> | gemini-2.0-flash-lite    | SelfCheckGPT   | +45.2% | -96% |
> |   | LLM as judge    | +18.1%    | -20%   |
> |    | **RvLLM**   | +24.9%   | -0.3%   |
> | deepseek-chat     | SelfCheckGPT   | +21.7%   | -87.0%  |
> |    | LLM as judge  | +9.6% | -65.2%  |
> |    | **RvLLM**     | +15.7%       | - 4.3%   |
>
> We observe that RvLLM achieves more stable performance across varied models, consistently maintaining a relatively low TNR drop while improving TPR high. In contrast, SelfCheckGPT often achieves the highest TPR increases simply because it tends to label nearly all outputs as “hallucinations” in our domain-specific settings, resulting in near-zero TNR. This behavior is expected, as SelfCheckGPT relies solely on output stability without incorporating domain knowledge. It does not encode or evaluate domain-specific rules, constraints, or context-sensitive reasoning, making it more suitable for open-domain generation tasks.
>
> As for LLM-as-judge method, we find it can achieve competitive TPR compared to RvLLM. However, its performance is also unreliable, with TPR gains typically accompanied by significant TNR drops. In contrast, RvLLM remains a better balance between TPR and TNR. This is because RvLLM leverages the LLM primarily for text understanding/reading, while delegating rule-based domain-related reasoning to an external reasoning engine, leading to more reliable and interpretable outcomes.
> We also include two additional comparison experiments corresponding to Case Studies 2 and 3.
>
> #### **For Case Study 2**:
> | verified model  | Method | TPR | TNR |
> | - | - | - | - |
> | Qwen-plus   | SelfCheckGPT    | 0  | 96.7%  |
> |   | LLM as judge     | 0    | 0 |
> |   | **RvLLM**    | 1    | 1  |
> | Qwen-turbo  | SelfCheckGPT   | 1    | 7.29%  |
> |  | LLM as judge      | 0    | 0 |
> |    | **RvLLM**    | 1    | 97.9%  |
> | gpt-4.1-mini  | SelfCheckGPT  | 1    |0  |
> |   | LLM as judge      | 1    | 1  |
> |   | **RvLLM**    | 88.9%    | 1 |
> | gemini-2.0-flash  | SelfCheckGPT   | 87.5%   | 0%  |
> |   | LLM as judge  | 1     | 1   |
> |    | **RvLLM**    | 1   | 1   |
> | gemini-2.0-flash-lite    | SelfCheckGPT  | 1 | 0 |
> |    | LLM as judge    | 1    | 1  |
> |  | **RvLLM**  | 1   | 1   |
> | deepseek-chat     | SelfCheckGPT   | 1   | 0  |
> |   | LLM as judge   | 1 | 1  |
> |   | **RvLLM**   | 1      | 93.2% |
>
> #### **For Case Study 3**:
>
> | verified model  | Method | TPR | TNR | Token Usage Per Task|
> | - | - | - | - | - |
> | Qwen-turbo   | SelfCheckGPT (NLI)   | 1    | 0  | 9850 |
> |   | SelfCheckGPT (Prompt)   | 1    | 0  | 100527|
> |   | LLM as judge      | 1    | 0 | 1466|
> |    | **RvLLM**    | 18.2%    | 94.4%  | 4805 |
> | gpt-4.1-mini  | SelfCheckGPT (NLI)   | 1    | 2%  | 10373 |
> |  | SelfCheckGPT (Prompt)   | 1    | 0  | 158504|
> |   | LLM as judge  | 1    | 0 |  2479 |
> |   | **RvLLM**   | 50%  | 84.2% | 5683 |
>
> Note that, in the current status of rebuttal, we do not provide more comparison studies to CheckEmbed mentioned in the reviews for the following several well-justified reasons:
>
> - First, both CheckEmbed and SelfCheckGPT share the same fundamental methodology of detecting hallucinations through output stability analysis, which assumes stable outputs are less likely to be hallucinated. While this approach proves effective for general NLP tasks like summarization, it demonstrates significant limitations when applied to our domain-specific constraint verification task. Most critically, stability-based methods are fundamentally incapable of: (1) reliably detecting constraint violations, or (2) identifying the precise sources of errors - both of which constitute essential requirements for our application.
>
> - Moreover, several practical considerations further motivated our decision. The original CheckEmbed paper reports performance comparable to or slightly worse than SelfCheckGPT and LLM-as-judge methods in stability detection. Additionally, CheckEmbed currently exists only as a non-peer-reviewed preprint on arXiv. From an implementation perspective, the tool's tightly coupled architecture presents substantial engineering challenges for integration, particularly given the already significant computational costs incurred (SelfCheckGPT alone required approximately 20 million tokens across our three benchmarks). These factors collectively made CheckEmbed adaptation impractical within the rebuttal timeframe.
>
> - Nevertheless, in direct response to the reviewer's constructive feedback, we have expanded our experimental analysis during the rebuttal phase to include comprehensive comparisons with SelfCheckGPT and LLM-as-judge (implemented via DeepSeek-Chat). We maintain that these comparisons adequately demonstrate our method's superior effectiveness against this class of stability-based detection approaches. Should the reviewer consider it essential, we would be pleased to conduct additional CheckEmbed experiments during the post-deadline discussion period, contingent upon the method's continued relevance to our evaluation framework.
>
> ### **Weakness W.2: Throughput and resource usage**
>
> We collect the info of our token usage (rounding to the nearest integer) on average for each single verification task in the three case studies as follows, where 1) Raw LLM denotes the tokens used for the initial query; and 2) the RvLLM denotes the tokens used by the LLM oracle in our verification framework.
>
> | Case Study 1 | Case Study 2 | Case Study 3 |
> | - | - | - |
> | 214 (Raw LLM) + 808 (RvLLM) | 36 (Raw LLM) + 1190 (RvLLM) | 1661 (Raw LLM) + 3669 (RvLLM) |
>
> ### **Weakness W.3: Generalization of ESL**
>
> We sincerely thank the reviewer for their valuable insights concerning the potential generalization of ESL to more expressive formal logic systems. The current implementation of ESL is limited to supporting basic logical and arithmetic predicates. Enhancing ESL's expressiveness constitutes a fundamental research challenge in LLM runtime verification – a direction we explicitly identify as important future work. We also give a discussion in the conclusion.
>
> However, we also acknowledge that monitoring cost can be higher given the more complicated syntax and logic, and that greater involvement from domain experts may be required to define fine-grained constraints. Indeed, the user-friendliness and the expressiveness also exist in a trade-off, which is indeed an interesting and meaningful research work.
>
> ### **Question Q.1: Mathematical guarantee**
>
> When the interpretation abstraction from natural language to ESL is correct (NL to ESL), RvLLM provides rigorous mathematical guarantees analogous to formal verification methods. Specifically, it offers complete verification coverage with respect to all domain-specific constraints encoded in ESL. As discussed in W.3, the accuracy of this interpretation abstraction is crucial - more complex logical syntax introduces greater challenges in maintaining abstraction fidelity. This fundamental trade-off between expressiveness and verifiability has motivated our current focus on simpler logical predicates and arithmetic relations in this initial framework.
>
> ### **Question Q.2: What if the domain knowledge is insufficient?**
>
> When knowledge is insufficient, some misconduct will not be verified as an error and marked as safe, i.e., a complete but unsound verification result.
>
> In practical applications, exhaustive and sufficient encoding of all relevant domain knowledge in a single iteration proves infeasible. A principal advantage of our framework lies in its ability to operate effectively without requiring complete domain knowledge a priori. Specifically, the system can perform verification based on currently specified constraints while supporting incremental integration of additional constraints as new domain knowledge is acquired. This design facilitates a scalable and pragmatic approach for progressively developing comprehensive domain specifications.
>
> Compared to approaches such as red teaming or dataset-based testing, our method supports incremental expansion of domain knowledge in a structured and reusable way. Moreover, the knowledge encoding process in our framework is typically a one-time effort, after which the constraints can be consistently reused across tasks and models, enabling systematic and ongoing refinement of model behavior.

---

> > ### Author Response · Authors · 2025-08-04
> >
> > Dear Reviewer, we truly appreciate your time and thoughtful consideration. We would like to kindly check whether we have addressed all your concerns. This paper is of great importance to us, and we have made a sincere effort to respond to each of your comments carefully. If possible, early feedback would be immensely helpful in allowing us to refine our responses and ensure that any remaining issues are fully resolved.

---

> > ### Comment · Reviewer_ywvD · 2025-08-06
> >
> > I thank the authors for the thorough rebuttal and the additional experiments. I am satisfied with your responses regarding W1 and W2, although these still represent limitations for the generalisation of ESL (W3).
> >
> > Regarding the verification results, I would argue that a verification method cannot be **complete but unsound**; it can only be sound but incomplete. A result that is truly complete must, by definition, also be sound. Therefore, I believe the notion of completeness here is subject to the extent of domain knowledge. If the underlying domain knowledge is insufficient or approximate, then the method may exhibit potential unsoundness. This is the reason why I believe caution is needed when using the term “verification.” The authors should make this distinction explicit in the paper, or, even better to avoid using it.

---

> > > ### Comment · Area_Chair_nwKN · 2025-08-06
> > >
> > > Thanks everyone for the engaged discussion.
> > >
> > > Verification methods can consider complete but unsound methods. As being sound requires soundness for all cases, any method that that always provides an answer but, in some cases, return incorrect answers would fall in this category. A typical case is a complete and sound method for a fragment of the logic that stills returns an answer out of the fragment but its meaningless. Complete without soundness might be seems trivial but knowing that the algorithm end is an important property. The language considered in the paper is used with arithmetic expressions. Adding arithmetic tend to turn out expressive logics into undecidable. However, there are useful methods or solvers dealing with arithmetic. For instance, SMT solvers.
> > >
> > > I understand the concern of the reviewer regarding the expressivity of the language. The authors might argue in favour by finding references on modelling with similar fragments. The caveat is the modification of forward chaining so it’s not clear for me now how to compare it.
> > >
> > > However, universally quantified formulas is a wide fragment.
> > >
> > > I invite the authors and the reviewer to consider the trade off having a more expressive language vs the computational cost at inference time.
> > >
> > > Lastly, the model checking research area usually assumes that that models are correct. The issue (the authors already touched this) is to be aware and offer ways to deal with the situation. I wonder if the modification for detecting inconsistency can help to realize model inconsistencies.
> > >
> > > It can also be considered a separated problem: checking if a formula actually matches a natural language sentence is a problem relevant for many neurosymbolic methods.

---

> ### Author Response · Authors · 2025-08-06
> **This is a response to both Reviewer ywvD and AC**
>
> We sincerely thank the reviewer and the chair for the in-depth discussion in response to our rebuttal, as well as for the thoughtful and valuable suggestions. **This is a response to both Reviewer ywvD and AC**
>
> ### **Regarding Verification Results**
>
> We fully acknowledge the importance of exercising caution when interpreting verification results. To avoid any potential misunderstandings, we are happy to include a more explicit and precise discussion of the verification terminology in the revised version. If the reviewer prefers, we are also open to replacing the term “runtime verification” with “runtime monitoring”. That said, we would prefer to retain “verification” to emphasize that our approach is grounded in formal specifications, thereby distinguishing it from conventional monitoring techniques.
>
> **Regarding the verification results, we agree with the chair’s view**. Additionally, we would like to offer a more intuitive explanation **from the perspective of our setting**:
>
> - Completeness, as mentioned in our response, refers to the ability of the verification process to identify all actual violations with respect to the encoded domain knowledge or constraints. In other words, if RvLLM reports a violation, it corresponds to a true violation by the model, which also reflects a genuine violation relative to the full knowledge base.
>
> - Soundness, in contrast, means that if RvLLM reports an output as safe (i.e., no violation), then the output is indeed safe. However, if the embedded domain knowledge is incomplete, RvLLM may produce unsound results, e.g., it might fail to detect certain unsafe behaviors because the relevant constraints were not encoded (as the chair explained, i.e., provides an answer but returns an incorrect answer)
>
> In this context, if we consider only the given domain knowledge, or assume that the domain knowledge is complete (which is possible in some domains with finite and well-defined constraints), then RvLLM can be considered both sound and complete, provided that the interpretation abstraction process is accurate.
>
> ### **Regarding Expressiveness**:
>
> We acknowledge that the current level of expressiveness may appear limited due to the simple syntactic structure (which currently resembles propositional logic). However, we would like to clarify several key aspects:
>
> 1. ESL is the first specification language that integrates both natural language and formal language, aiming to strike a balance between "user-friendliness & flexibility" and "formal rigor". This integration enables users to flexibly define their own properties, with the flexibility largely stemming from ESL’s natural language binding.
>
> 2. We currently support basic arithmetic operations in the syntax, with the actual computations handled by Python during execution.
>
> 3. Although ESL’s surface syntax may appear propositional, semantically we operate over a fragment of first-order logic, specifically, **prenex normal forms** (as we noted in Section 3.1). That is, each ESL rule is interpreted as a prenex normal form during the interpretation phase.
>
> The reason we chose not to incorporate explicit quantifier operators at the syntactic level is also explained in the final paragraph of Section 3.1. In short, this design decision simplifies both syntax and interpretation by shifting the quantifier semantics entirely to the runtime. In practice, during the interpretation stage, we extract all relevant instantiations of each predicate from the LLM context. A single ESL rule will thus be expanded into a set of propositional rules, effectively capturing universal quantification through enumeration over these instances.
>
> As for further extending expressiveness, such as incorporating temporal constraints (e.g., via temporal logic or Signal Temporal Logic), we are actively exploring this direction now as future work. We believe that this work provides a solid foundation for such extensions.
>
> Finally, as we have discussed, striking the right balance between expressiveness, runtime efficiency, and user accessibility remains an ongoing challenge. A more complex language could lead to decreased accuracy in model abstraction for the specification rules, potentially undermining the effectiveness (and also efficiency) of runtime verification.

---

> > ### Author Response · Authors · 2025-08-06
> > **(Continued)**
> >
> > **Regarding the chair’s question: "I wonder if the modification for detecting inconsistency can help to realize the model inconsistencies”**. If we understood the question correctly, our interpretation is as follows: If the notion of inconsistency—under a specific modeling setting—can be formally encoded as some ESL rules, then RvLLM can be configured to first check whether the model abstracted violates these inconsistency constraints. If no inconsistencies are detected, the system can then proceed to the second stage: verifying whether the model satisfies the user-defined properties of actual interest.
> >
> > Moreover, as the chair rightly pointed out in the last, fully automating the transformation from natural language to formal language is still a highly non-trivial task. For more complex settings, we believe that runtime language refinement strategies may offer a promising path forward in addressing this challenge.

---

> > ### Comment · Reviewer_ywvD · 2025-08-08
> >
> > Thanks for the discussion. Perhaps “runtime validation” would be a more suitable term, or you could state very clearly in your revised version that the results are subject to the assumption of certain domain knowledge, and highlight the potential risk of unsoundness; also, for the limitations of the generalisation of ESL.
> >
> > I will maintain my original score.

---

### Decision · Program_Chairs · 2025-09-17

**Decision:**

Accept (poster)

**Comment:**

The paper introduces a lightweight logic to be used for runtime detection of inconsistencies. The logic is a fragment of first-order logic, where the formulas are of the form DNF \implies CNF, and the predicates P(X,Y) are associated with natural language expressions. The LLM is queried to populate the grounded predicates P(a,b) to assumed to be true, and a variation of forward chaining is used to detect inconsistencies. Inconsistencies are used to reject decoding paths in the LLM.

The reviewers praised the equilibrium between a simple language suitable for modelling and an effective mechanism, demonstrated using Singapore traffic regulation.

Some of the weaknesses mentioned included that the logic might be too simple to represent complex domains, and the cost of expressing the rules in logic. The author's response focuses on the value of achieving higher robustness at the price of modelling the rules.

The reviewers commented on the lack of comparison with verification using LLMs. The authors provided experimental results that convinced the reviewers.

Other reviewers pointed out how, as the number of rules increases, it might be harder to detect mistakes. I consider this slightly out of scope, but the authors perform an experiment adding additional rules manually and showing the different kinds of errors that might appear.

The reviewers didn’t discuss in depth the technical details of the logic and the inference mechanism, probably due to a lack of background. This is NeurIPS and is increasingly more challenging to find researchers familiar with the large body of work on using logic for AI tasks and model checking.

In my opinion, the algorithm is focused on detecting inconsistencies in tractable time, and dealing with the open domain assumption (absence of predicate P(a) does not mean we can assume -P(a)). The authors might want to look into intuitionistic logics, as they don’t include the axiom $P \lor \-P$.

The comment on Signal Temporal Logic (STL) is, honestly, irrelevant. STL is an extension of LTL, linear temporal logic. The ESL language only uses universal quantifiers. (STL is not mentioned in the manuscript, so there is nothing to fix there).

An improved version of this work should look into Datalog and other extensions of Horn clauses. Here is a recent relevant reference:
Martin Babka, Tomáš Balyo, Ondřej Čepek, Štefan Gurský, Petr Kučera, Václav Vlček,
"Complexity issues related to propagation completeness",
Artificial Intelligence, 2013, https://doi.org/10.1016/j.artint.2013.07.006.

Meanwhile, here is a technical detail: The transformation mentioned in L218, section 4.2, obtains horn clauses from the original ESL formulas. A detail is mentioned without full justification. For all rules with CNF of the right side, a rule is added to conclude each literal, by adding the condition that all the others are false. That is for a role $t \implies p \lor q \lor r$, rules of this form are generated: $t \land \-p \land \-q \implies r$

This transformation is equivalent to “unit propagation”! https://en.wikipedia.org/wiki/Unit_propagation
The most basic inference used in SAT solvers, and is part of the original paper on DPLL.

However, that mechanism is sound for detecting inconsistencies, but is not complete. For that, consider the SAT problem: check whether a CNF formula $\phi = \phi_1 \land \dots \land \phi_n$, where each $\phi$ is a disjunction, is satisfiable. That problem is the prototypical NP-complete problem. That formula can be converted into an ESL by just using DNF = true. We don’t even need predicate variables. In this case, ESL might not detect inconsistencies, but the formula might be unsatisfiable. (If you want a concrete example, look for trivial encodings of the https://en.wikipedia.org/wiki/Pigeonhole_principle)
However, from the purely algorithmic point of view, reducing the inference to what we can get from unit propagation makes it efficient and can detect non-trivial inconsistencies.

The informal discussion of the logic is acceptable for a paper in NeurIPS, at least nowadays. It would have been harder to accept in AAAI or in general AI journals like AIJ or JAIR.

I strongly recommend that the authors add a related work subsection connecting with relevant work, and not imply in any way that the algorithm is complete. Please do not say or imply in any way that “ESL is the first specification language that integrates both natural language and formal language”. Natural language interpretations have a long tradition in AI. It was popular in the 80s to mix expert systems with NLP.